# Bacterial meningitis in the early postnatal mouse studied at single-cell resolution

**Jie Wang[1,2], Amir Rattner[1], Jeremy Nathans[1,2,3,4]***

[1]Department of Molecular Biology and Genetics, Johns Hopkins University School of Medicine, Baltimore, United States; [2]Howard Hughes Medical Institute, Johns Hopkins University School of Medicine, Baltimore, United States; [3]Department of Neuroscience, Johns Hopkins University School of Medicine, Baltimore, United States; [4]Department of Ophthalmology, Johns Hopkins University School of Medicine, Baltimore, United States

*For correspondence:
jnathans@jhmi.edu

**Competing interest:** The authors declare that no competing interests exist.

**Abstract** Bacterial meningitis is a major cause of morbidity and mortality, especially among infants and the elderly. Here, we study mice to assess the response of each of the major meningeal cell types to early postnatal *E. coli* infection using single nucleus RNA sequencing (snRNAseq), immunostaining, and genetic and pharamacologic perturbations of immune cells and immune signaling. Flatmounts of the dissected leptomeninges and dura were used to facilitiate high-quality confocal imaging and quantification of cell abundances and morphologies. Upon infection, the major meningeal cell types – including endothelial cells (ECs), macrophages, and fibroblasts – exhibit distinctive changes in their transcriptomes. Additionally, ECs in the leptomeninges redistribute CLDN5 and PECAM1, and leptomeningeal capillaries exhibit foci with reduced blood-brain barrier integrity. The vascular response to infection appears to be largely driven by TLR4 signaling, as determined by the nearly identical responses induced by infection and LPS administration and by the blunted response to infection in *Tlr4⁻/⁻* mice. Interestingly, knocking out *Ccr2*, encoding a major chemoattractant for monocytes, or acute depletion of leptomeningeal macrophages, following intracebroventricular injection of liposomal clodronate, had little or no effect on the response of leptomeningeal ECs to *E. coli* infection. Taken together, these data imply that EC responses to infection are largely driven by the intrinsic EC response to LPS.

## Editor's evaluation

This study presents valuable findings on the changes in immune cell populations and stromal cells occurring at the CNS borders in a neonatal bacterial meningitis model, focusing on fibroblasts, macrophages, and endothelial cells. The study provides a solid snRNA-seq dataset and high-quality immune fluorescence images of dissected brain border regions, that will be useful for the community. These observations and datasets will be of interest to the neuro-immunology community.

## Introduction

The brain and spinal cord are protected, both physically and immunologically, by the meninges, a multi-layered tissue that occupies the space between the CNS parenchyma and the surrounding bone (Figure 1A; *Coles et al., 2017a*; *Weller et al., 2018*). Starting from the surface of the brain and moving toward the skin, the meninges consists of: (1) the pia, a thin and semi-permeable layer of cells that allows passage of small molecules and proteins between the CNS parenchyma and the cerebro-spinal fluid (CSF); (2) the sub-arachnoid space, a highly vascularized region containing fibroblasts and immune cells that is filled with CSF and supported by a web of trabeculae; (3) the arachnoid, including

an outer epithelial barrier layer that serves as the outer boundary of the CSF-accessible space; and (4) the dura, a fibrous layer containing draining sinuses (veins), lymphatics, fibroblasts, and immune cells. Layers 1–3 together comprise the leptomeninges.

The meninges hosts a diverse collection of immune cells, including macrophages [referred to as barrier-associated macrophages (BAMs) or CNS-associated macrophages (CAMs)], monocytes, innate lymphoid cells (ILCs), T-cells, B-cells, dendritic cells, and mast cells. Recent single-cell RNA sequencing (scRNAseq) and immuno-phenotyping have revealed distinctive layer-specific types and abundances of these immune cells (*Rua and McGavern, 2018*; *Alves de Lima et al., 2020*). Macrophages are especially abundant, and they can be divided into several molecularly distinct classes that are characterized by different half-lives and capacities for self-renewal (*Mrdjen et al., 2018*; *Kierdorf et al., 2019*; *Van Hove et al., 2019*; *Masuda et al., 2022*). Layer specific diversity is also observed among meningeal fibroblasts, with molecularly distinctive pial, arachnoid, and perivascular fibroblasts, as well as two types of dural fibroblasts (*DeSisto et al., 2020*; *Derk et al., 2021*).

A wide variety of CNS injuries and disease processes involve the meninges, including stroke, traumatic brain injury, neuroinflammatory conditions such as multiple sclerosis, neurodegenerative diseases, and infections (*Derk et al., 2021*; *Alves de Lima et al., 2020*). Disease and injury responses have been observed among meningeal immune cells, fibroblasts, and vasculature (*Rua and McGavern, 2018*; *Derk et al., 2021*). Recent experiments with animal models imply that some of these changes play a causal role in injury or disease pathology. For example, in mouse models of stroke, genetic ablation of meningeal mast cells reduces infarct size and brain swelling (*Arac et al., 2014*).

The oldest and best-established pathophysiologic role for the meninges is as a site of bacterial, fungal, or viral infection (*Uiterwijk and Koehler, 2012*; *Williamson et al., 2017*; *Kohil et al., 2021*). Bacterial meningitis is most common among young children and the elderly, and it is generally initiated by a blood-borne infection (*Ku et al., 2015*; *McGill et al., 2016*). The annual incidence of bacterial meningitis ranges from ~2 per 100,000 people in Western Europe and North America to 100–1000 per 100,000 people in the Sahel region of Africa (*Ku et al., 2015*; *GBD 2016 Meningitis Collaborators, 2018*). In Western Europe and North America, mortality from bacterial meningitis is 10–20%, with >30% of survivors experiencing residual neurologic defects (*McGill et al., 2016*; *Eisen et al., 2022*). The molecules and mechanisms that mediate bacterial adhesion to and invasion of the meningeal vasculature are objects of active investigation (*Coureuil et al., 2017*).

Meningitis in the neonatal period, when the immune system is immature, typically results from bacterial infection during or immediately prior to delivery. Bacteria, most commonly Group B *Streptococci* and *E. coli*, are introduced into the bloodstream through breaks in the infant's skin or via intra-amniotic infection (*Gaschignard et al., 2011*; *Shane et al., 2017*). The incidence of bacterial meningitis in neonates is ~0.3 per 1000 live births in developed countries and ~4 per 1000 live births in less developed countries (*Ku et al., 2015*). Among survivors of neonatal meningitis, 20–70% (the number depending on the country) are left with long-lasting neurologic sequelae, including hearing loss, epilepsy, and learning and/or behavioral disabilities (*Peltola et al., 2021*).

In the present work, we describe the molecular and cellular responses of meningeal cells in a mouse model of neonatal *E. coli* meningitis. Responses to infection were observed in every major meningeal cell type, including endothelial cells (ECs), macrophages, and fibroblasts. We have also used genetic and pharmacologic perturbations of immune cells and pathways to explore communication networks responsible for this complex multi-cellular response.

## Results

### Single nucleus sequencing and flatmount imaging of the mouse meninges

Our point of departure in studying the murine meninges was to utilize a simple protocol for dissecting the leptomeninges and the dura free from adjacent tissues. When the skull and brain are separated in the absence of fixation, the natural cleavage plane is between the leptomeninges and dura (*Figure 1A*). The leptomeninges can then be peeled from the brain surface and the dura can be peeled from the inner surface of the skull. In our hands, the separation of the unfixed leptomeninges from the brain parenchyma works best with tissue from young mice: it is efficient with tissue harvested at postnatal day (P)6 but, as noted by *Van Hove et al., 2019* and confirmed by us, it fails with adult

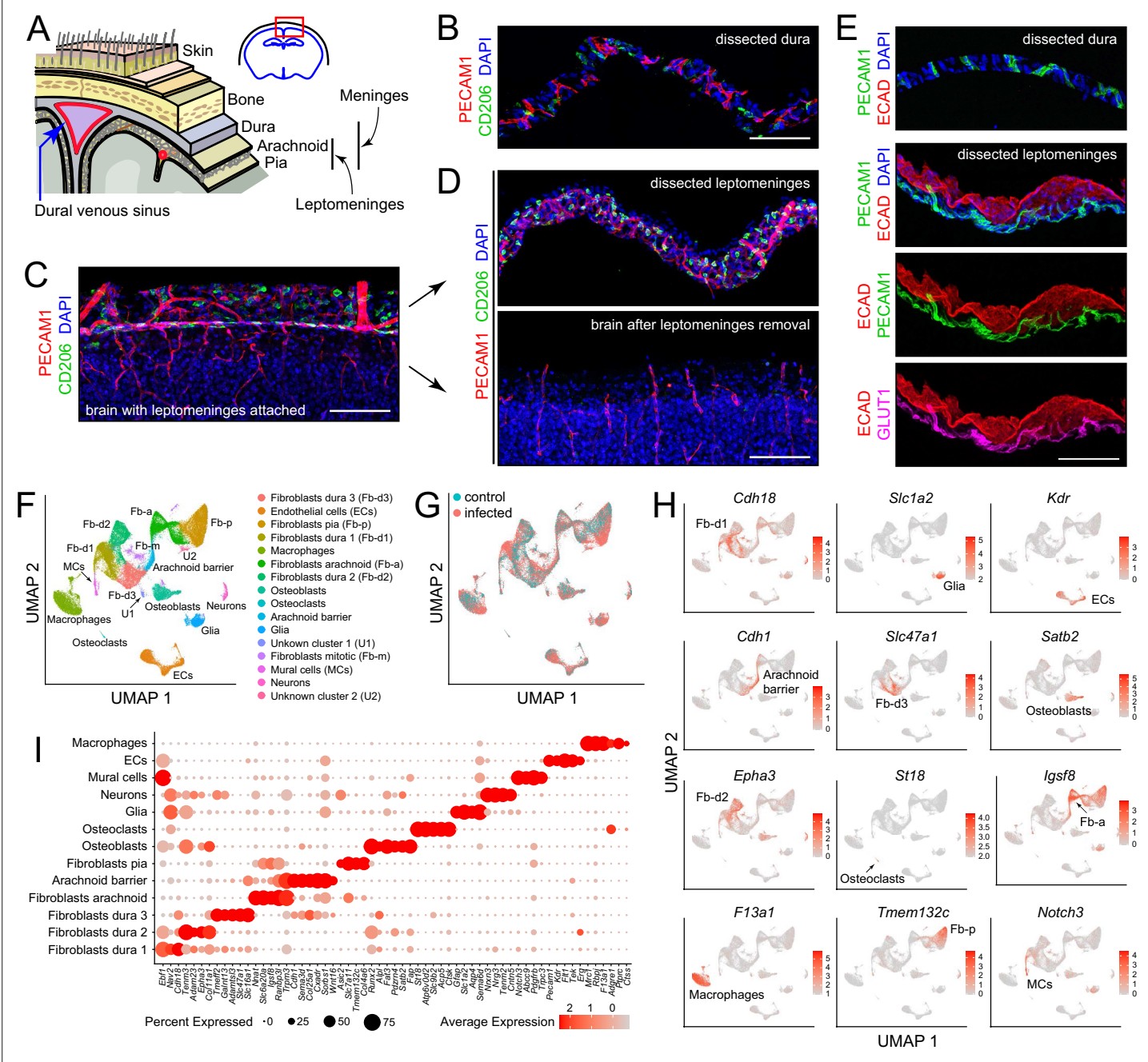

**Figure 1.** Dissection and single nucleus (sn) RNAseq of mouse meninges. (**A**) Diagram of the tissue layers between brain and skin, corresponding to the red rectangle in the coronal section through the brain and skull (upper right). (**B**) Cross-section of dissected dura stained for PECAM1 (endothelial cells) and CD206 (macrophages). (**C**) Coronal section through cortex (lower) and overlying leptomeninges (upper) stained for PECAM1 and CD206. (**D**) Cross-scetion of the isolated leptomeninges (upper panel) and the denuded brain (lower) stained for PECAM1 and CD206. (**E**) Dissected dura and leptomeninges stained for PECAM1 and ECAD (arachnoid barrier epithelium); the leptomeninges is also stained for GLUT1 (a BBB marker; bottom panel). (**F**) UMAP plot of combined control and infected meninges snRNAseq datasets with cell clusters differentially colored and labeled. The macrophage cluster includes a small upward extension that represents monocytes and monocyte-derived cells. (**G**) UMAP plots of separated control and infected meninges snRNAseq datasets. (**H**) UMAP plots, as in panel (**F**) showing transcripts that are highly enriched in each of 12 cell clusters (labels in each panel). (**I**) Dot plot showing some of the transcript abundances that most clearly discriminate among major meningeal cell types, as well as contaminating neurons and glia. Scale bars: B-E, 100 um. All tissue and data in this and other figures are from P6 mice. The immunostaining and histochemical probes in this and other figures are indicated adjacent to the corresponding panel(s), with lettering color-coded to match the corresponding fluorescent color.

The online version of this article includes the following figure supplement(s) for figure 1:

*Figure 1 continued on next page*

*Figure 1 continued*

**Figure supplement 1.** Pairwise Pearson correlations among snRNAseq datasets.

**Figure supplement 2.** Flatmount images of control (i.e. uninfected) leptomeninges and dura.

**Figure supplement 3.** Characteristics of the P5-P6 model of *E. coli* meningitis.

tissue. Separation of the leptomeninges is also more readily accomplished if the brain is first chilled in ice-cold PBS for several minutes. Immunostaining of the isolated leptomeninges, the isolated dura, and the denuded brain reveals ECs (expressing PECAM-1/CD31) and macrophages (expressing CD206/MRC1) in both leptomeninges and dura (*Figure 1B–D*). The arachnoid epithelium expresses E-cadherin (ECAD). Leptomeningeal ECs, but not dura ECs, express GLUT1, a blood-brain barrier (BBB) marker (*Figure 1E*). These and all other analyses in this study were conducted at P6.

For transcriptome analyses at cellular resolution, we sought to obtain as representative a sampling of cell types as possible and to minimize RNA synthesis or degradation after dissecting the dura and leptomeninges. Therefore, we avoided enzymatic tissue dissociation followed by single cell analysis and, instead, purified nuclei following tissue homogenization. The meninges were obtained from P6 mice, either without infection ("control") or following a single subcutaneous injection of *E. coli* K1 at P5 ('infected'), as described more fully in the next section. Each sequencing library, two control and three infected, was produced from a single mouse and consisted of the pooled leptomeninges plus dura tissues. All experiments, except for those shown in Figure 6C-F and 7, were conducted with FVB/NJ mice. Single nucleus (sn)RNAseq data were obtained from 14,356 and 34,585 nuclei from control and infected mice, respectively, with a mean of 1,320 transcripts sequenced per nucleus, using the 10 X Genomics Chromium platform (*Supplementary file 1*). Pairwise Pearson correlations among the five snRNAseq datasets (two control and three infected) shows correlations of 0.98–1.00 within the same group and 0.90–0.93 between infected vs. control groups (*Figure 1—figure supplement 1*).

Fourteen principal cell clusters were identified with Seurat, and their identities were assigned by immunostaining and with reference to published data, as seen in the Uniform Manifold Approximation and Projection (UMAP) plots in *Figure 1F and H* (*Supplementary file 2*). One cluster, labeled 'macrophages', consists largely of macrophages but also encompasses other immune cells, as described in detail below. Two clusters represent neurons and glia, presumably brain contaminants. One small cluster derives from mitotic fibroblasts, and two small clusters (U1 and U2) are unidentified. Strikingly, five large clusters represent fibroblasts – one pial, one arachnoid, and three dural, as determined by a comparison with published data on meningeal fibroblasts (*DeSisto et al., 2020*; *Derk et al., 2021*). A dot plot of 61 transcripts that exhibit cell-type-specific patterns of enrichment supports these cell cluster assignments and also illustrates a pattern of partial overlap in gene expression among the five fibroblast clusters (*Figure 1I*). Comparing UMAP plots of control and infected datasets reveals shifts in the positions of the immune cell and fibroblast clusters with infection (*Figure 1G*).

Flatmounts of the isolated leptomeninges and dura permit confocal imaging across the full depth of each of these tissues (*Figure 1—figure supplement 2*). Additionally, the dura can be imaged as a flatmount while it remains attached to the inner surface of the skull, although this arrangement reduces tissue access to antibody and washing solutions. Leptomeninges flatmounts show a high density of macrophages marked by (1) co-expression of macrophage markers CD206 and LYVE1, which localize to distinct cytoplasmic/surface compartments (*Figure 1—figure supplement 2A–C*). These cells, as well as other non-macrophage immune cells, also express SPI1/PU.1, which localizes to the nucleus, and CD45/PTPRC, a general marker for hematopoietic cells (*Figure 1—figure supplement 2A–C*). Flatmounts of the peripheral dura (i.e. outside the sinuses) show elongated perivascular macrophages expressing CD206 and LYVE1, as well as additional immune cells expressing CD45 (*Figure 1—figure supplement 2D*).

The vasculature can be visualized in leptomeninges and dura flatmounts by immunostaining for PECAM-1, and, in leptomeninges flatmounts, by immunostaining for tight junction markers Occludin (OCLN), Zonula occludens-1 (ZO-1), and Claudin-5 (CLDN5) (*Figure 1—figure supplement 2C* and below). In dura flatmounts with the bone attached, perivascular fibroblasts express FOXP2, and osteoblasts express SATB2 (*Figure 1—figure supplement 2D*).

## The bacterial meningitis model

To model human neonatal meningitis, we inoculated P5 mice with *E. coli* strain RFP-RS218 (O18:K1:H7). This strain is a clinical isolate from the cerebrospinal fluid (CSF) of a neonate with meningitis and it has been derivatized by the addition of a plasmid expressing red fluorescent protein (RFP) to facilitate visualization of *E. coli* cells in tissue. P5 was chosen as the age of inoculation to approximate the degree of maturity of the neonatal human immune system (*Holsapple et al., 2003*; *Kuper et al., 2016*; *Park et al., 2020*). Presumably, the 9-month human gestation compared to the 19–20 day mouse gestation explains the relatively greater developmental maturity of the human immune system compared to the mouse immune system at birth.

In each experiment, a litter of P5 mice were randomly divided into two groups that were either subcutaneously injected in the back with $1.2 \times 10^5$ CFU of *E. coli* RFP-RS218 in 20 μL PBS or not injected. Twenty-two hours later (at P6), the mice were sacrificed and tissues analyzed. One day after *E. coli* inoculation, the mice appeared lethargic and they stopped gaining weight (*Figure 1—figure supplement 3A*). Any inoculated mice that were not sacrificed died within two days of the inoculation (*Figure 1—figure supplement 3B*).

At sacrifice, 22 hr after subcutaneous inoculation, *E. coli* cells were typically observed in a patchy distribution in the leptomeninges and dura, and at much sparser density within the brain (*Figure 2A–D*). The number of *E. coli* per unit area in flatmounts of the dura sinus region was, on average, ~20-fold greater than the number of *E. coli* in the same area in flatmounts of the leptomeninges, reflecting, in part, the several fold greater depth of the dura sinus (*Figure 1—figure supplement 3C*, left plot). The spatial heterogeneity in *E. coli* accumulation within the meninges, together with animal-to-animal variation in the severity of infection, likely accounts for some degree of variability in the cellular alterations associated with infection. To address this issue, representative images were chosen for the figures and for quantification. All images and data in the present study were obtained at P6, and all quantifications of flatmount images used Z-stacks that span the full thickness of the tissue.

## Overview of the responses of meningeal cells to infection

In comparing control vs. infected datasets for each of the principal cell clusters, scatter plots encompassing all transcripts reveal relatively few changes in the contaminating neuron and glia clusters and many more changes in each of the principal meningeal clusters, with roughly equal numbers of up- and down-regulated transcripts (*Figure 2—figure supplement 1* and *Supplementary file 3*). A comparison of non-immune meningeal cells (excluding contaminating non-meningeal cells [neurons, glia, osteoblasts, and osteoclasts]) that was limited to transcripts with a log2-fold change equal to or greater than 2.5 in control vs. infection conditions in any one or more of these cell types shows that dural and leptomeningeal ECs form one cluster and dural and leptomeningeal fibroblasts and arachnoid barrier cells form a second cluster (*Figure 2—figure supplement 2*). The following transcripts are increased broadly across cell types: (1) general stress response genes Metallothionien-1 (*Mt1*), Metallothionien-2 (*Mt2*), lipocalins Apolipoprotein D (*Apod*) and Lipocalin2 (*Lcn2*), (2) Lipopolysaccharide (LPS) binding protein (*Lbp*, which presents LPS to TLR4), (3) Serum amyloid A3 (*Saa3*, an acute phase protein induced by inflammation), and (4) Ceruloplasmin (*Cp*, a secreted copper-binding protein that is also induced by inflammation).

Dot plots that include all of the cell types in the snRNAseq dataset were generated to visualize representative examples of the various patterns of altered transcript abundance with infection (*Figure 2—figure supplement 3*). Several transcripts – for example, *Kiz*, *Slc39a14*, *Ap4e1*, and *Cp* – are up-regulated across nearly all clusters (*Figure 2—figure supplement 3A*). However, for most transcripts that exhibit abundances changes with infection, those changes were limited to smaller subsets of cell types, for example, the down-regulation of *Col14a1*, *Col8a1*, and *Slc4a10* in dural fibroblasts, the down-regulation of *Igsf8*, *Tmtc4*, and *Zfp536* in arachnoid and pial fibroblasts and arachnoid barrier cells, and the up-regulation of *Cxcl2* in macrophages (*Figure 2—figure supplement 3A and B*). Hierarchical clustering in a gene set enrichment analysis (GSEA) with control vs. infected samples shows prominent induction of inflammatory responses ('complement', 'interferon gamma response', 'IL6 JAK STAT3 signaling', 'TNFA signaling via NFKB') across all meningeal cell types. ECs and arachnoid barrier cells show a reduction in apical barrier (i.e. tight junction) transcripts, but this reduction did not reach statistical significance (*Figure 2—figure supplement 4*).

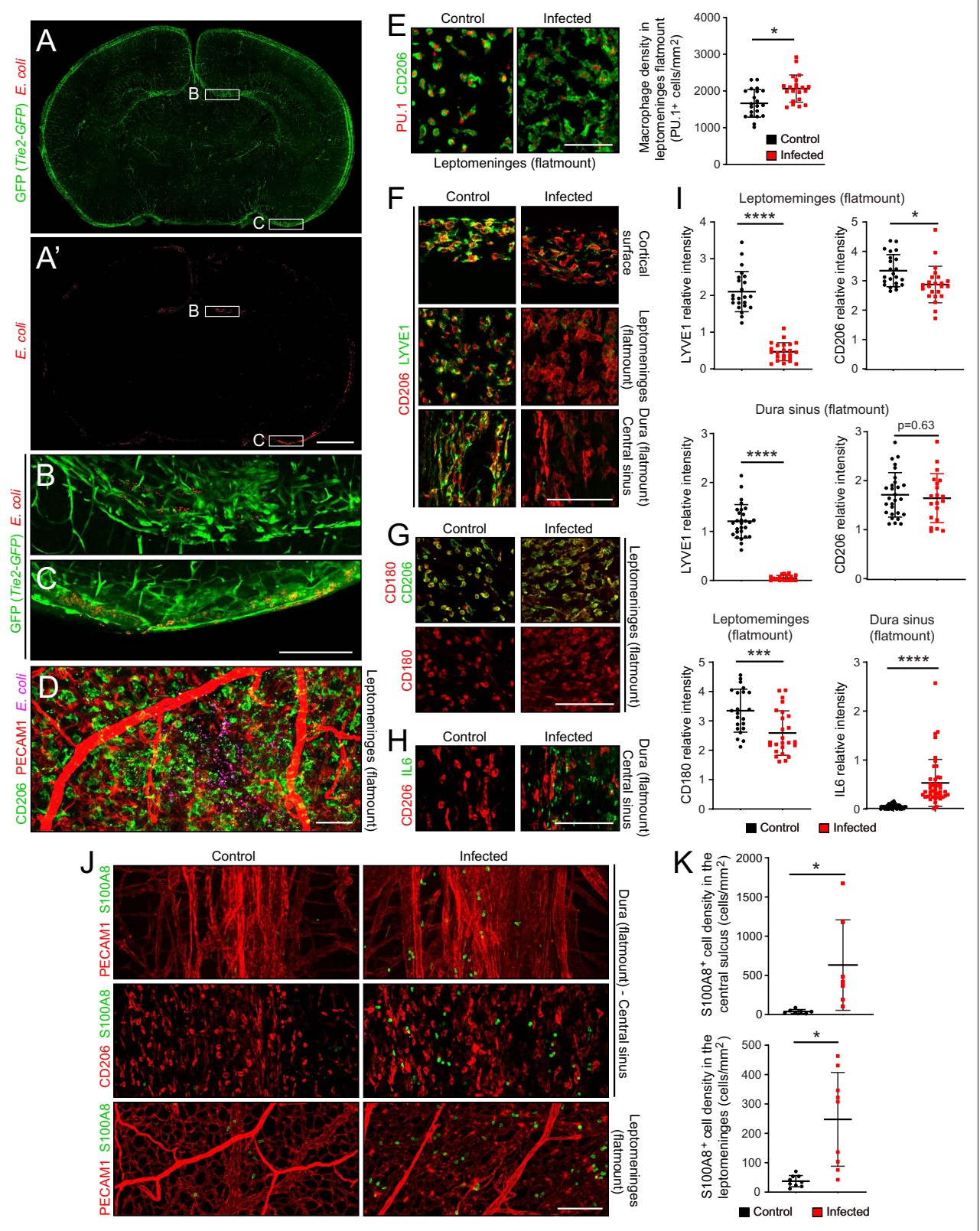

**Figure 2.** The *E. coli* meningitis model and some immune cell responses. (**A–C**) Coronal sections of P6 brain with leptomeninges, 1 day after a subcutaneous injection of 1.2×10⁵ RFP-expressing *E. coli* K1. Regions within the rectangles labeled (**B**) and (**C**) are enlarged below. The *Tie2-GFP* transgene is expressed in ECs. (**D**) Leptomeninges flatmount showing scattered *E. coli* (RFP; false colored magenta). (**E**) Leptomeningeal macrophages, visualized with nuclear immunostaining for transcription factor PU.1 and cytoplasmic staining for CD206, show cytoplasmic enlargement upon

*Figure 2 continued on next page*

*Figure 2 continued*

infection (left), but only a small increase in cell number (right). (**F**) Infection leads to a selective reduction in LYVE1 immunostaining, and little or no change in CD206 immunostaining, in macrophages in the leptomeninges and dura. (**G**) Infection leads to little or no change in CD180 and CD206 immunostaining in macrophages in the leptomeninges. (**H**) IL6 increases in dural fibroblasts in the central sinus. (**I**) Quantification of images in F-H, in arbitrary units. Each point in this and other quantifications of immunofluorescent data represents the analysis of a single Z-stacked confocal image that encompasses the full depth of the tissue (leptomeninges or dura), unless noted otherwise. (**J and K**) Increase in cells immunostained for S100A8 in the leptomeninges and dura of infected mice. All infected tissue and data in this and other figures are from P6 mice that had been infected 22 hr earlier. Scale bars: A and A', 500 μm; B-I, 100 μm. In this and all other figures showing quantification: (1) unless stated otherwise, each symbol in the immunofluorescent quantifications represent a single confocal image, (2) the bars show mean ± SD; (3) the number of mice used for each sample are listed in *Supplementary file 4*; (4) the Wilcoxon rank sum test was used to measure statistical significance, except for *Figure 5D and G*, in which the sample size is too small and the student t-test was used instead; and (5) abbreviations are: n.s., not significant (i.e. $p>0.05$); *, $p<0.05$; **, $p<0.01$; ***, $p<0.001$; ****, $p<0.0001$.

The online version of this article includes the following figure supplement(s) for figure 2:

**Figure supplement 1.** Scatterplots for the major meningeal cell types comparing snRNAseq transcript abundances in control vs. infected mice.

**Figure supplement 2.** Heatmap of non-immune meningeal cells showing all transcripts with log2-fold change greater than 2.5 in control vs. infection conditions in any one or more of the listed cell types.

**Figure supplement 3.** Dot plot showing some of the transcript abundance patterns that distinguish control vs. infected meninges, plotted by cell type.

**Figure supplement 4.** Gene set enrichment analysis (GSEA) for individual cell types in a comparison of control vs. infected meninges.

## Responses of meningeal immune cells to infection

Despite the presence of *E. coli*, the density of macrophages in the leptomeninges, which are normally present at 1500–2000 cells per mm$^2$, increased by only ~20% (*Figure 2E*; for all image quantifications, the number of mice in each sample are listed in *Supplementary file 4*). Whether this increase reflects in situ proliferation, ingress from other tissue compartments [e.g. blood and/or skull bone marrow (*Herisson et al., 2018*; *Cugurra et al., 2021*)], or a combination of the two, remains to be determined. However, macrophage morphology changed dramatically with infection, from small and rounded to large and irregularly shaped (*Figure 2E*). Infection also led to a reduction in LYVE1 abundance, but little or no change in CD180 or CD206 abundance, in macrophages (*Figure 2F, G and I*). In the dura of infected mice, IL6 was induced in fibroblasts, with the cell type assignment determined by the observed increase in fibroblast-specific *Il6* transcript abundance in the snRNAseq datasets (*Figure 2H and I*, and *Figure 2—figure supplement 3A*). In both the dura and leptomeninges, the number of cells expressing S100A8, a marker for monocytes and immature macrophages, increased ~10-fold (*Figure 2J and K*).

For a more comprehensive assessment of the immune response to infection, we further parsed the immune and immune-related clusters into microglia (a brain contaminant), innate lymphoid cells/T cells (ILC/T), osteoclasts (a contaminant from the skull), monocytes (MCs) and monocyte-derived cells, resident macrophages (MPs; subdivided by CCL2 expression), and inflammatory macrophages (subdivided by IL1 receptor type 1 [IL1R1] expression; *Figure 3A–C , and D* lower panel). Cell clusters were assigned with reference to published data, as summarized in *Supplementary file 2*. The designation of macrophage clusters as 'resident' or 'inflammatory' reflects their locations in the UMAP plots (i.e. their gene expression profiles), the former corresponding to UMAP locations occupied by macrophages in the control meninges and the latter corresponding to UMAP locations occupied by macrophages that are specific for the infected meninges (compare *Figure 3A and F*). Importantly, these designations refer only to patterns of gene expression and are agnostic as to the origins and migratory histories of the macrophage clusters. The division of resident macrophages into CCL2$^-$ and CCL2$^+$ subtypes is based on the differential expression of multiple genes, six of which are included in the dot plot in *Figure 3B*, with these six plus an additional 24 also included in the dot plot in *Figure 3—figure supplement 1A*.

Although the relative abundances of the principal meningeal cell types did not change with infection, as judged by counting nuclei in the five snRNAseq libraries (*Figure 3D* upper panel), parsing the individual immune cell types revealed an ~twofold decrease in the abundance of CCL2$^-$ resident macrophages, a > 10-fold increase in inflammatory macrophages (IL1R1$^+$ and IL1R1$^-$), and a several-fold increase in the abundance of monocytes or monocyte-derived cells (*Figure 3D*, lower panel). While these experiments do not distinguish between changes in gene expression patterns among resident immune cells versus the ingress of circulating immune cells, the changes in macrophage

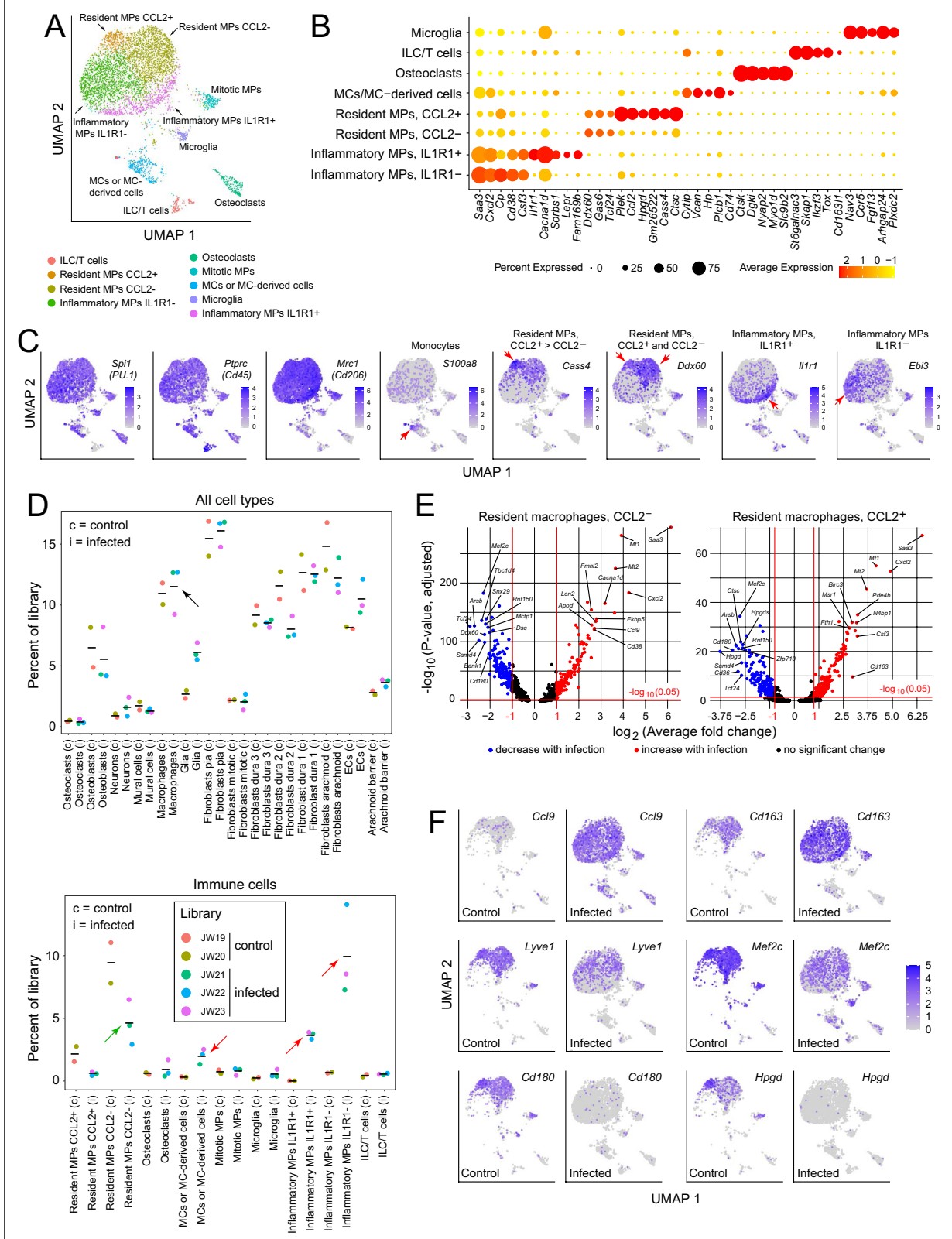

**Figure 3.** Immune subtypes and their responses to infection. (**A**) snRNAseq UMAP plot for immune cells from combined control and infected meninges. (**B**) Dot plot showing some of the transcript abundances that most clearly distinguish among meningeal immune cells. (**C**) UMAP plots, as in panel (**A**) showing eight transcripts that are expressed by all (left three panels) or by distinct subsets (right five panels) of macrophage subtypes and macrophage-like cells. Red arrows highlight regions within the UMAP clusters that correspond to distinct cell types, as defined in panel

*Figure 3 continued on next page*

*Figure 3 continued*

(**A**). (**D**) Comparing the number of nuclei in control (**c**) vs. infected (**i**) snRNAseq datasets across all meningeal cell types (upper panel) and across immune cells (bottom panel). The fraction of cells comprising the general category 'macrophages' shows no change with infection (black arrow in upper panel). However, the lower panel shows that several macrophage subsets decrease (green arrow) or increase (red arrows) in abundance with infection. (**E**) Volcano plots for CCL2⁻ and CCL2⁺ macrophages showing control vs. infected snRNAseq transcript abundances (see *Supplementary file 5*). (**F**) UMAP plots, as in (**A**), comparing control vs. infected snRNAseq for six genes.

The online version of this article includes the following figure supplement(s) for figure 3:

**Figure supplement 1.** Comparisons between CCL2⁻ and CCL2⁺ macrophages.

**Figure supplement 2.** Leptomeningeal and dura fibroblast markers and fibroblast responses to infection.

**Figure supplement 3.** Collagen transcripts in individual meningeal cell types from control vs. infected mice.

**Figure supplement 4.** SLC transporter transcripts in individual meningeal cell types from control vs. infected mice.

abundances could be largely explained if ~50% of the CCL2⁻ resident macrophages present prior to infection converted to inflammatory macrophages in response to infection. Among genes that are either up- or down-regulated in CCL2⁻ or CCL2⁺ resident macrophages, several dozen show greater than 5-fold changes in abundance (*Figure 3E* and *Supplementary file 5*). As seen in *Figure 3—figure supplement 1B*, CCL2⁻ and CCL2⁺ macrophages show nearly identical changes among the transcripts with the greatest increases in abundance with infection, and somewhat greater variability among transcripts with the greatest decreases in abundance with infection.

The UMAP plots in *Figure 3F* compare the expression levels of six genes in control vs. infected immune cells, and they illustrate the appearance of the inflammatory macrophage pattern of gene expression specifically in the infected meninges, represented by the lower ~50% of the macrophage cluster (see also *Figure 3A*). *Figure 3F* also illustrates the diversity of macrophage transcript changes with infection, with dramatic up-regulation of *Ccl9*, modest up-regulation of *Cd163*, modest down-regulation of *Lyve1* and *Mef2c*, and dramatic down-regulation of *Cd180* and *Hpgd*. Interestingly, by immunostaining, CD180 levels showed little change at this time point (22 hr post-infection; *Figure 2G and I*), suggestive of a long protein half-life, whereas LYVE1 levels showed a large reduction (*Figure 2F and I*), suggestive of post-transcriptional as well as transcriptional down-regulation. These data are consistent with a model in which infection promotes the appearance of new macrophages 'states', as defined by novel patterns of gene expression that are distinct from those of resting macrophages.

## Responses of meningeal fibroblasts to infection

Fibroblasts constitute the most abundant cell type in the leptomeninges and dura (*Figure 1F*, *Supplementary file 1*, and *Figure 3—figure supplement 2*), and each of the five meningeal fibroblast subtypes shows numerous changes in transcript abundances in response to infection (*Figure 2—figure supplements 1–4* and *Supplementary file 3*). Here, we highlight two gene families, collagens and SLC transporters, in which multiple family members show expression changes in control vs. infected meningeal fibroblasts. Transcripts coding for multiple collagen subtypes are down-regulated by infection: among the 50 members of the collagen gene family, 25 show detectable expression in the meninges by snRNAseq and 19/25 are down-regulated but only 2/25 are up-regulated (*Figure 3—figure supplement 3*). Two examples are shown in the UMAP plots in *Figure 3—figure supplement 2*: *Col14a1* is down-regulated in type 1 and 2 dural fibroblasts, and *Col25a1* is down-regulated in arachnoid barrier cells and type 3 dural fibroblasts. Immunostaining of the dura for COL14A1 did not reveal significant changes one day after infection, likely reflecting the slow turnover of mature extracellular matrix collagen (*Jackson and Heininger, 1975*; *Last et al., 1989*).

Similarly, multiple transcripts coding for SLC transporters are down-regulated in meningeal fibroblasts. Of the ~350 members of the mouse *Slc* gene family with detectable expression in meningeal cells, 37 show infection-dependent changes in transcript abundance in one or more meningeal cell types by snRNAseq, with 21/37 down-regulated and 6/37 up-regulated by $\log_2$-fold>0.25 following infection (*Figure 3—figure supplement 4*). The UMAP plots in *Figure 3—figure supplement 1B* show down-regulation of *Slc16a1* in dura fibroblasts. *Slc39a14*, which codes for a divalent metal transporter, is unusual in its substantial up-regulation with infection (*Figure 3—figure supplement 4B*).

In contrast to the pattern of down-regulation among the majority of *Col* and *Slc* transcripts, across the full transcriptome similar numbers of transcripts are up- and down-regulated by infection in each

of the major meningeal cell types (*Figure 3—figure supplement 1*). Among fibroblasts, examples of transcripts that are up-regulated include *Alk* in arachnoid fibroblasts, *Scara5* in pial fibroblasts, *Nrg3* in type 1 dura fibroblasts, *Camk4* in type 2 dura fibroblasts, and *Lbp* in type 3 dura fibroblasts (*Figure 3—figure supplement 2B*). As noted above, infection up-regulates IL6 protein and *Il6* transcripts in dura fibroblasts (*Figure 2H and I*, and *Figure 3—figure supplement 2B*).

## Responses of meningeal vasculature to infection

In the context of bacterial meningitis, meningeal ECs serve as both a portal of entry for bacteria and immune cells and as a site of pathologically increased vascular permeability (*Kim et al., 1997*; *Barichello et al., 2013*; *Coureuil et al., 2017*). To explore the meningeal EC response to infection, we first parsed the EC UMAP into clusters derived from leptomeninges, dura, and arterial ECs based on the following markers: BBB markers *Cldn5*, *Lef1*, *Slc2a1*, and *Slc7a1* for leptomeningeal ECs vs. non-BBB marker *Plvap* for dural ECs; and arterial markers *Bmx* and *Fbln5* (*Figure 4A and C*; *Supplementary files 2 and 6*). Dural ECs were further divided into Von Willebrand Factor (VWF) expressing and non-expressing subclasses (*Figure 4A and C*). Immunostaining of leptomeninges and dura flatmounts for CLDN5, SLC2A1/GLUT1, and PLVAP confirmed the cluster assignment of leptomeningeal ECs vs. dura ECs, and immunostaining for smooth muscle actin (SMA) was used to distinguish arteries vs. veins histologically.

Infection leads to an expansion in the size of the leptomeninges and dura EC clusters (*Figure 4A and E*), indicative of increased heterogeneity in transcriptome content, with little or no change in the proportions of the different EC subtypes (*Figure 4B*). Multiple transcriptome changes distinguish the infection responses of different EC clusters, including down-regulation of transcripts coding for amino acid transporters SLC7A1 and SLC7A5 in leptomeningeal ECs and down-regulation of transcripts coding for plasmalemma vesicle associated protein (PLVAP; a marker of high permeability vasculature) and the mechanosensory channel PIEZO2 in dural ECs (*Figure 4C–E*). Transcripts coding for VEGFR2/KDR are down-regulated in leptomeningeal and dural ECs (*Figure 4C–E*).

In leptomeninges flatmounts, infection produces mislocalization and clustering of CLDN5 and PECAM1, disorganized capillary morphology, and an expansion of the area covered by capillaries (*Figure 5A–C*). Immunoblotting of leptomeningeal proteins shows a modest reduction in the mean level of CLDN5, but this trend did not reach statistical significance due to the relatively high animal-to-animal variability in the infected group (*Figure 5D*). Functionally, there are patchy deficiencies in vascular barrier integrity following infection, with a spatial distribution that closely matches the clustering of CLDN5 and PECAM1, as revealed by extravasation of Sulfo-NHS-biotin, a low molecular weight intravascular tracer (*Figure 5A–C*). The same infection-associated vascular phenotypes were also seen one day after an intraperitoneal (IP) injection of 10 mg/kg lipopolysaccharide (LPS), a potent activator of the innate immune response to gram-negative bacteria such as *E. coli* (*Figure 5A–C*).

Two well-studied signaling pathways are known to control CNS vascular permeability: VEGF and WNT. Consistent with the down-regulation of *Vegfr2/Kdr* transcripts seen in infected ECs by snRNAseq (*Figure 4C–E*), VEGFR2 immunostaining is also greatly reduced in the leptomeninges (*Figure 5E and H*). As VEGFR2 is the principal receptor for VEGF signaling in ECs, its down-regulation suggests that the enhanced vascular permeability associated with bacterial infection is not caused by increased VEGF signaling, a known mechanism for increasing vascular permeability (*Senger et al., 1983*; *Roberts and Palade, 1995*).

WNT signaling in CNS vasculature maintains the blood-brain barrier (BBB) and it is both mediated by and up-regulates the transcription factor LEF1, which binds to target genes in combination with beta-catenin (*Sabbagh et al., 2018*). In control leptomeninges flatmounts, LEF1 accumulates in vein and capillary EC nuclei with minimal accumulation in arterial ECs (*Figure 5F*, left panels). Strikingly, in infected leptomeninges flatmounts, LEF1 immunostaining in ECs is greatly reduced, but it persists in many non-ECs (*Figure 5F*, right panels; and *Figure 5H*). By immunoblotting, total leptomeningeal LEF1 levels are reduced by ~25%, a change that was not statistically significant (*Figure 5G*). The right panels of *Figure 5F* shows reductions in CLDN5 and PECAM1 specifically in leptomeningeal veins, which appear as 'shadows' in the flatmount images. With infection, a reduction is also seen for ERG, a pan EC transcription factor (*Figure 5F and H*). These data are consistent with a model in which infection leads to reduced BBB integrity in the leptomeningeal vasculature, at least in part, by reducing WNT signaling in ECs.

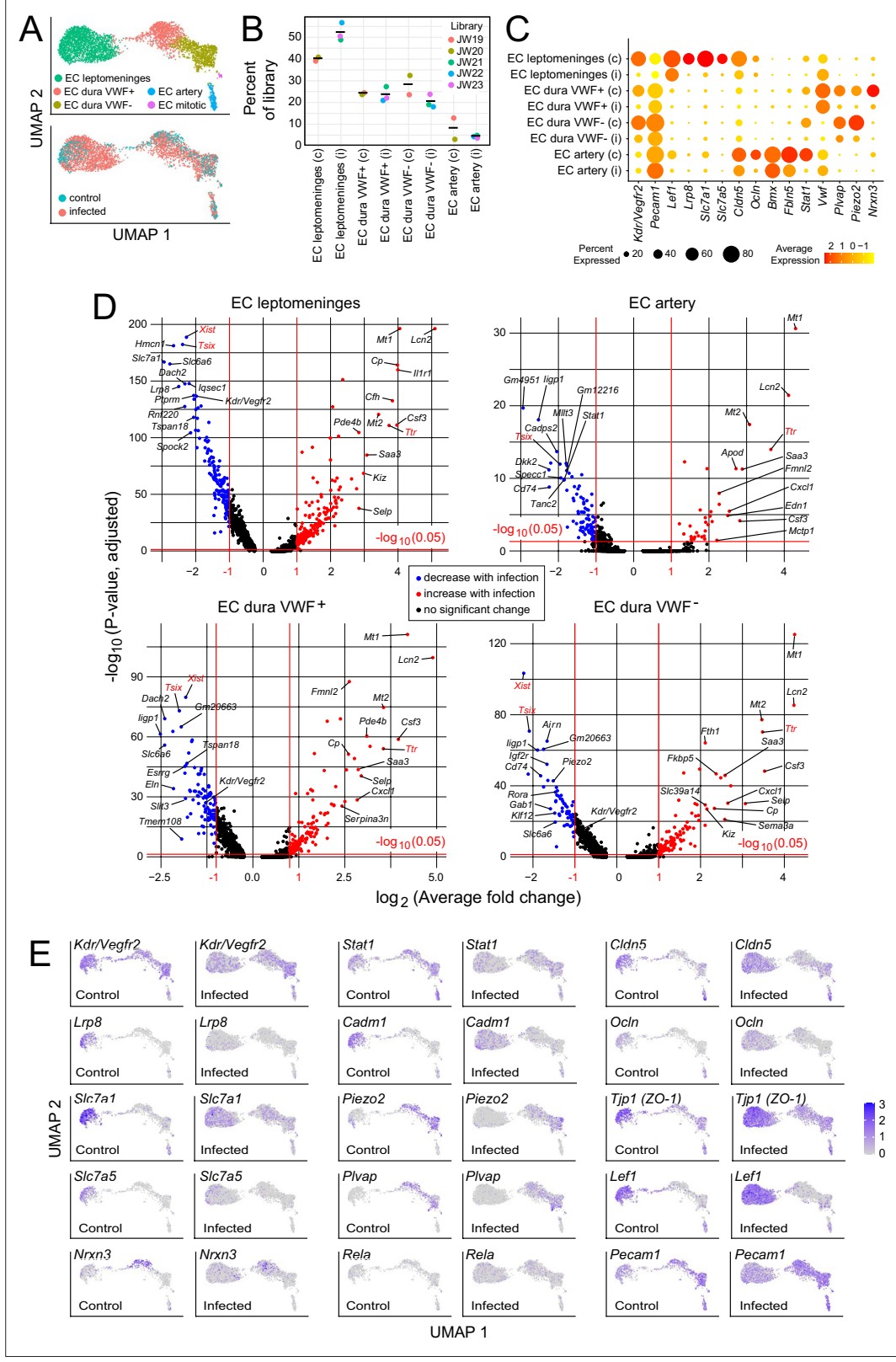

**Figure 4.** Changes in EC gene expression with infection in the leptomeninges and dura. (**A**) snRNAseq UMAP plots for ECs from combined control and infected meninges. (**B**) The number of nuclei from different EC subtypes is consistent across the five snRNAseq libraries (control: JW19, JW20; infected: JW21-JW23). (**C**) Dot plot showing changes in transcript abundances in EC subtypes in control (**c**) vs. infected (**i**) snRNAseq datasets. (**D**) Volcano

*Figure 4 continued on next page*

*Figure 4 continued*

plots for four EC subtypes showing control vs. infected snRNAseq transcript abundances. *Xist* and *Tsix* transcripts, referable to sex differences among embryos, are marked in red. *Ttr* transcripts, also marked in red, likely represent contamination from choroid plexus RNA and are present in 2/3 infected snRNAseq samples (see ***Supplementary file 6***). (**E**) UMAP plots, as in (**A**), comparing control vs. infected snRNAseq for 15 genes.

To assess the effects of infection on the organization of dura ECs, we took advantage of the large and unusually straight veins that occupy the dural sinuses adjacent to the skull's sutures. Venous ECs throughout the body typically exhibit elongated nuclei that are aligned with the long axis of the vein, and therefore also with the direction of blood flow. With infection, this alignment is diminished in dural veins (***Figure 6A and B***), suggesting a general effect of infection on cytoskeletal organization within venous ECs. No assessment of permeability was made for the dura vasculature because the dura resides outside of the BBB territory (delimited by the arachnoid epithelial barrier) and, therefore, it exhibits high permeability in the control state.

Microbial products, such as LPS, and any of a wide variety of cytokines could directly or indirectly alter EC structure and gene expression. As multiple immunologic stimuli are known to converge on the NF kappaB pathway and as the GSEA analysis implied that this pathway was induced upon infection (***Figure 2—figure supplement 4***), we assessed NF kappaB signaling by generating reporter mice in which five tandem repeats of a canonical NF kappaB response element were inserted upstream of a minimal promoter to drive expression of a nuclear localized and 3xHA epitope-tagged tandem dimer Tomato from the *Rosa26* locus (nls-tdT-3xHA; ***Figure 6C***). The parental version of this mouse line has the additional feature that a loxP-transcription stop-loxP cassette separates the promoter and the nls-tdT-3xHA coding region, permitting cell-type specific read-out of NF kappaB signaling when crossed to a cell-type specific Cre transgene or knock-in allele. For the present analyses, we have used germline Cre-recombination to generate an allele that is predicted to be permissive for reporter expression in any cell type.

In both dura and leptomeninges, NF kappaB reporter expression was induced by infection almost exclusively in ECs, with reporter positive EC nuclei increasing from ~1% to~20% in the dura and from ~1% to~5% in the leptomeninges. EC nuclei were identified based on ERG immunostaining (***Figure 6D–F***). As the the *Rosa26* locus is generally permissive for expression in most, if not all, cell types, it was surprising that NF kappaB reporter expression was largely restricted to ECs. A second surprising feature was the cell-to-cell heterogeneity in EC expression, with reporter expressing ECs adjacent to non-expressing ECs in an apparently random pattern. The latter observation suggests substantial cell-to-cell heterogeneity in meningeal EC responses to infection, consistent with the observed broadening of infected EC clusters in the UMAP plots in ***Figure 4A and E***.

## Genetic and pharmacologic perturbations of the immune response

To explore the role of specific immune pathways in the vascular changes associated with infection, we applied the *E. coli* infection paradigm to mice with null mutations in (1) *Tlr4*, the gene coding for the innate immune system's LPS receptor, or (2) *Ccr2*, the gene coding for one of the receptors for monocyte chemoattractant protein-1 (MCP1/CCL2), a chemokine that recruits immune cells to sites of infection (***Figure 7***). *Tlr4* is expressed widely among cell types within the meninges, whereas *Ccr2* is expressed predominantly in monocytes and monocyte-derived cells (***Figure 7—figure supplement 1***; in the dot plot of all meningeal cells shown in ***Figure 7—figure supplement 1B***, monocytes are included in the macrophage cluster). For this experiment, the WT comparator strain is C57BL/6 J, which matches the background of the *Tlr4*[-/-] and *Ccr2*[-/-] mice. The EC response to infection is milder in C57BL/6 J mice compared to FVB/NJ mice. At one day post-infection, the number of *E. coli* in leptomeninges flatmounts did not differ significantly between C57 WT control and *Ccr2*[-/-] mice, whereas the number of *E. coli* in leptomeninges flatmounts was increased ~twofold in *Tlr4*[-/-] mice, albeit with substantial scatter in the data (***Figure 1—figure supplement 3C***, center plot).

In response to infection, the leptomeningeal vasculature of *Tlr4*[-/-] mice showed minimal changes in the distribution of CLDN5 and very few regions of Sulfo-NHS-biotin leakage. In contrast, *Ccr2*[-/-] mice showed a clumpy redistribution of CLDN5 and localized regions of Sulfo-NHS-biotin leakage, much like the C57BL/6 J WT control (***Figure 7A***). Quantifying the area occupied by the leptomeningeal vasculature and the extent of Sulfo-NHS-biotin leakage confirmed this visual impression (***Figure 7B***).

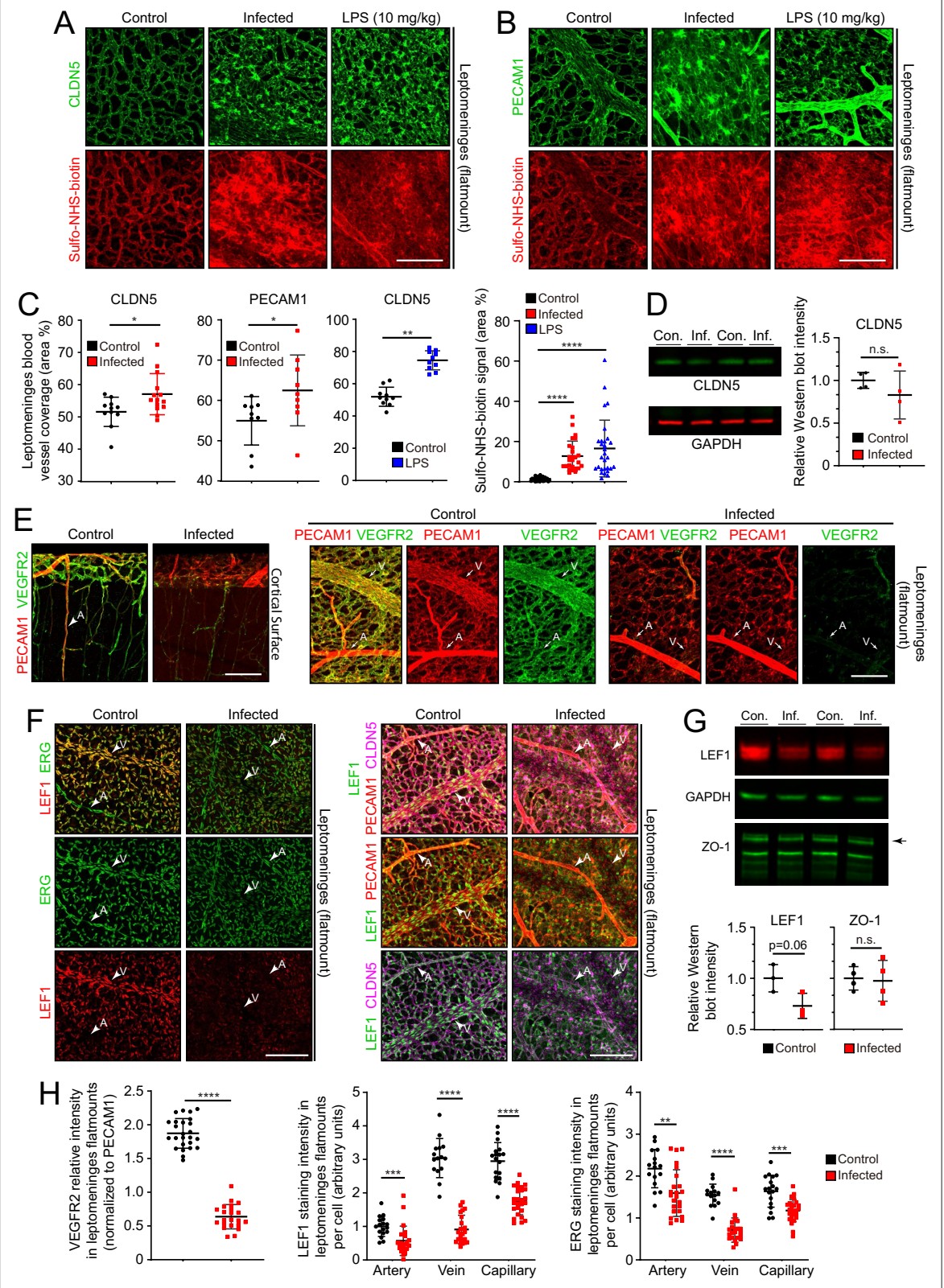

**Figure 5.** Changes in EC morphology and EC protein abundance and localization with infection. (**A**) CLDN5 localization and Sulfo-NHS-biotin leakage in leptomeningeal vasculature following infection or LPS administration. (**B**) PECAM1 localization and Sulfo-NHS-biotin leakage in leptomeningeal vasculature following infection or LPS administration. (**C**) Infection or 10 mg/kg LPS treatment increases the area covered by vasculature in flatmounts of leptomeninges. (**D**) Immunoblotting shows a modest, but not statistically significant, reduction in CLDN5 level relative to GAPDH level in the

*Figure 5 continued on next page*

*Figure 5 continued*

leptomeninges with infection (n=4 independent experiments). (**E**) Reduced KDR (VEGFR2) immunstaining in leptomeninges ECs with infection. (**F**) Reduced EC nuclear LEF1 immunostaining in capillaries and veins, and reduced PECAM1 and CLDN5 staining in veins in the leptomeninges with infection. (**G**) Immunoblotting shows a modest reduction in LEF1 level and no change in ZO-1 level relative to GAPDH level in the leptomeninges with infection (n=3 independent experiments for LEF1 and n=4 independent experiments for ZO-1). (**H**) Quantification of images in E and F. For (**D**) and (**G**), the statistical signficance was calculated using the student's t-test because the Wilcoxon rank sum test cannot be used on such small sample sizes. Scale bars: A and B, 100 μm; E and F, 100 μm.

In control (i.e., uninfected) *Tlr4*$^{-/-}$ and *Ccr2*$^{-/-}$ mice, the density of leptomeningeal macrophages (as quantified by PU.1 immunostaining) was, respectively,~65% and~90% of the WT value, and this density rose by ~50% in infected *Tlr4*$^{-/-}$ mice, but showed little or no change in infected *Ccr2*$^{-/-}$ or C57 WT mice (**Figure 7C**). Similar results were obtained by quantifying CD206 immunostaining, with *Ccr2*$^{-/-}$ leptomeninges showing ~80% of the C57 WT value (**Figure 7—figure supplement 2** compares PU.1 and CD206 quantification). The lower baseline number of macrophages in the *Tlr4*$^{-/-}$ leptomeninges compared to WT could reflect a modestly higher level of macrophage proliferation in the WT driven by a basal level of constitutive TLR4 signaling, perhaps in response to LPS from the developing gut and/or skin microbiomes or from adult feces in the cage. The lower baseline number of macrophages in the *Ccr2*$^{-/-}$ leptomeninges compared to WT could reflect reduced monocyte ingress.

As a complementary approach to gene inactivation, we used a single intra-cerebroventricular (ICV) injection of clodronate-containing liposomes to acutely and selectively eliminate leptomeningeal macrophages 2 days before *E. coli* infection (**Figure 8A and B**). The number of *E. coli* in the leptomeninges did not differ significantly between mice that received control vs. clodronate liposomes (**Figure 1—figure supplement 3C**, right plot). Surprisingly, eliminating leptomeningeal macrophages had little or no effect on the infection-dependent redistribution of CLDN5 in leptomeningeal ECs, leakage of Sulfo-NHS-biotin, the increase in the area occupied by leptomeningeal vasculature, or the fractional increase in the number of ECs showing induction of the NF kappaB reporter (**Figure 8B-D**).

The principal differences between responses of mice receiving control liposomes vs. clodronate liposomes were the modestly higher overall levels (i.e. both baseline and infected) following clodronate treatment of (1) the leptomeningeal blood vessel area (**Figure 8D**, second plot) and (2) the number of leptomeningeal ECs with NF kappaB reporter activation (expressed as HA+/ERG +ECs; **Figure 8D** third plot). These modest effects of clodronate treatment might represent inflammatory/stress responses that arise from the death of large numbers of leptomeningeal macrophages and the accompanying release of bioactive substances. In support of the earlier inference that relatively few macrophages migrate into the leptomeninges 1 day after infection, the data in **Figure 8C and D** show that after clodronate depletion of CNS macrophages at P3, the number of macrophages in the leptomeninges at P6 increases only modestly following infection.

Taken together, these experiments, together with the LPS treatment experiment (**Figure 5A–C**), show that (1) LPS stimulation of TLR4 signaling plays a central role in the response of the leptomeningeal vasculature to infection (CLDN5 and PECAM1 redistribution, vessel swelling, and leakage), and (2) this vascular response is largely independent of leptomeningeal macrophages, by far the most abundant immune cells in the leptomeninges.

## Discussion

The present study defines the responses of cells in the mouse leptomeninges and dura to bacterial meningitis in the early postnatal period. At this age, the immaturity of the adaptive immune system and the rapidity of infection imply that the host response depends largely, and perhaps exclusively, on the innate immune system. In response to bacterial infection, all of the major meningeal cell types – including ECs, macrophages, and fibroblasts – exhibit large and distinctive changes in their transcriptomes. In addition, ECs in leptomeningeal capillaries redistribute CLDN5 and PECAM1, leptomeningeal capillaries become enlarged and disorganized and they exhibit foci of reduced BBB integrity, and ECs in leptomeningeal capillaries and veins lose nuclear LEF1. These capillary responses to infection appear to be largely driven by TLR4 signaling, as determined by the response to LPS administration and by the blunting of these responses to bacterial infection in the absence of TLR4. This simple and robust model of bacterial meningitis in infancy should prove useful in dissecting mechanisms of

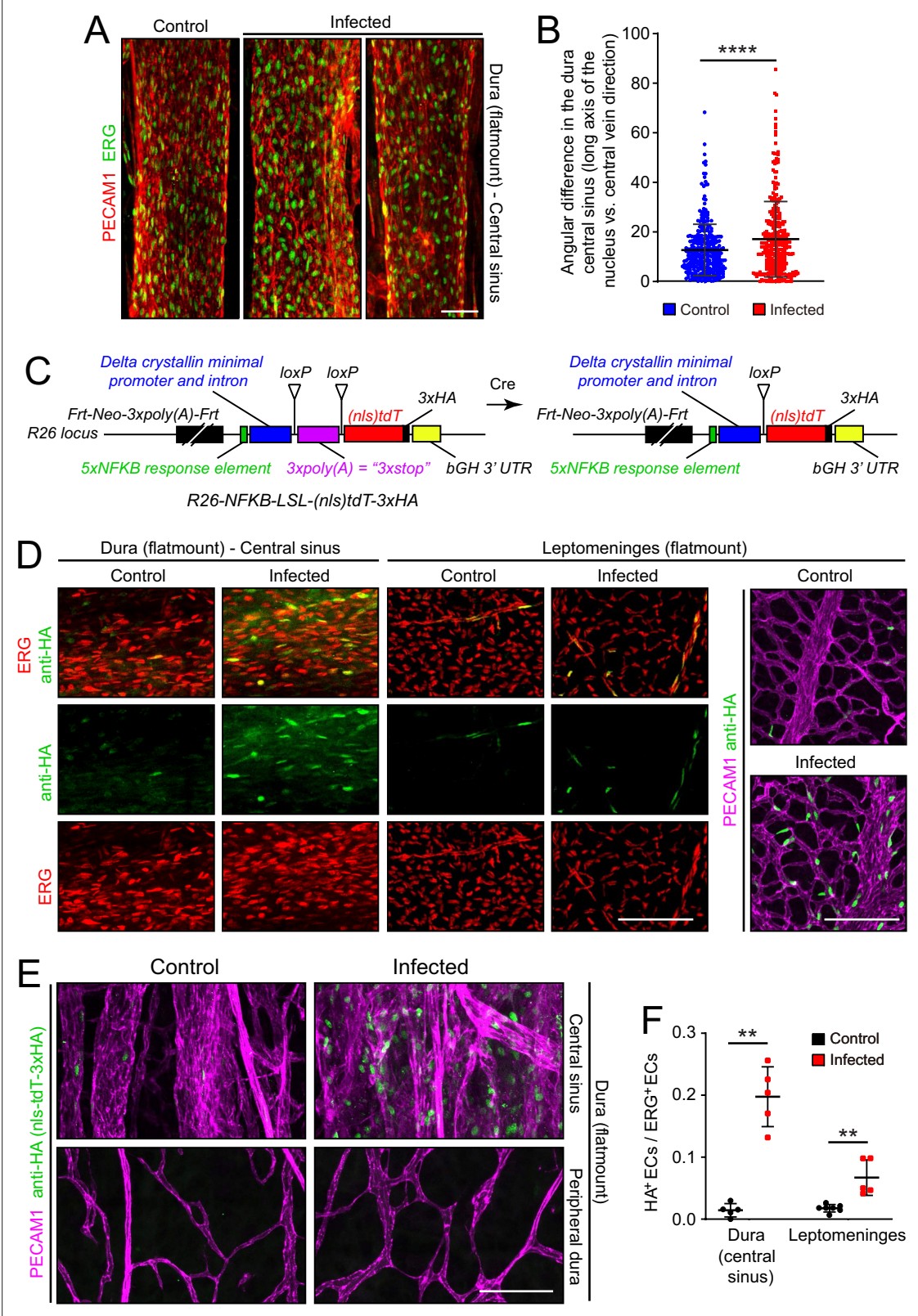

**Figure 6.** Infection causes disorganization of EC nuclear orientation in the dural venous sinus, and an increase in NF-kappa B signaling in ECs in the leptomeninges and dura. (**A**) EC nuclei, visualized with ERG immunostaining, in the large vein of the central sinus. (**B**) Quantifying the orientation of the long axis of EC nuclei in the large vein of the central sinus, as shown in (**A**). Each data point represents one nucleus. (**C**) Structure of the NF-kappa B reporter before (left) and after (right) Cre-mediated recombination that removes a loxP-transcription stop-loxP (LSL) cassette. NF-kappa B

*Figure 6 continued on next page*

*Figure 6 continued*

reporter activation leads to expression of nls-tdT-3xHA. (**D**) Infection increases expression of the NF-kappa B reporter in a subset of ECs in the dura and leptomeninges, as determined by immunostaining for HA. (**E**) In the dura, NF-kappa B reporter activation is observed in both ECs and non-ECs in the central sinus, but the NF-kappa B reporter is not activated in the peripheral dura. (**F**) Quantification of NF-kappa B reporter activation in ECs in the central sinus of the dura and in the leptomeninges. Scale bars: A, 50 µm; D and E, 100 µm.

pathophysiology. In the future, it would be interesting to modify the mouse model by including antibiotic treatment at different times after infection to parallel the clinical course of treated meningitis and to explore the neurologic sequelae that are commonly seen in human survivors of bacterial meningitis.

This study has several limitations. First, the analyses focused on the response at 22 hr post-infection and earlier and later events have not been studied. Second, the movements, if any, of immune cells between compartments have not been studied, and therefore it is unclear whether some of the changes in transcriptome profiles should be ascribed to changes in cell state – that is the same cells as were present prior to infection, but with an altered transcriptional program – or the ingress of immune cells from non-meningeal pools. Interestingly, previous cell tracing analyses have shown that the pool of myeloid cells that supplies the meninges includes cells that are locally sequestered in the skull bones (*Herisson et al., 2018*; *Cugurra et al., 2021*). The modest increases in macrophage density in the leptomeninges following infection, with or without clodronate treatment (*Figures 2E, 8C and D*), imply a correspondingly modest influx of cells from non-meningeal reservoirs. Third, most of the transcriptomic changes are, at present, of unknown functional significance, and, more specifically, the clinical significance of the observed cellular and molecular changes is also not clear. Future work will be aimed at defining the physiologic significance of the observed transcriptome changes. For example, increases or decreases in the abundance of particular SLC transcripts (*Figure 3—figure supplement 4*) imply corresponding changes in the transmembrane movement of a defined set of small molecules.

Comparisons with previously published single cell (sc) RNAseq data from mouse meninges emphasize the challenges of correlating data from tissues harvested at different ages, prepared by different methods, and subject to undefined batch effects. For example, in the present study we have divided dura fibroblasts into three clusters (*Figure 1F, H and I*), with clusters Fb-d1 and/or Fb-d3 likely derived from embryonic day (E)14 dura fibroblast clusters M4-1 and M4-2, as defined by *DeSisto et al., 2020*. Based on a comparison to scRNAseq of mouse coronal sutures dissected at E15.5 and E17.5 (*Farmer et al., 2021*), cluster Fb-d2 likely corresponds to Farmer et al.'s MG2 (*Matn4* +and *Nppc*+, and occupying the outer dura) and Fb-d3 likely corresponds to Farmer et al.'s MG3 (*Matn4*- and *Nppc*+, and occupying the inner dura). Additional challenges attend comparisons between human and mouse meninges datasets, as scRNAseq of adult human dura reveals an even more complex landscape, with subdivision of dural fibroblasts into either six or 14 clusters, depending on the analysis method (*Wang et al., 2022*). Comparisons between P6 and adult macrophage subtypes in the mouse meninges, the latter defined by *Van Hove et al., 2019*, are challenging as adult meningeal macrophages were subdivided based on MHC class II (e.g. H2-Aa) transcript levels, which are uniformly low in P6 meningeal macrophages.

The role of LPS in mediating BBB breakdown has been intensively studied, primarily in the context of the vasculature within the brain parenchyma (*Wispelwey et al., 1988*; *Banks et al., 2015*). In the brain, LPS treatment leads to a reduction in EC tight junctions secondary to a decrease in tight junction protein abundance and changes in tight junction protein localization (*Peng et al., 2021*). In multiple cell types, TLR4 signaling (i.e. LPS-induced signaling) activates the NF-kappa B pathway, and that connection presumably accounts for the NF-kappa B reporter activation in meningeal ECs observed here. Although the infection-induced down-regulation of transcripts coding for cell-cell junction proteins in ECs, arachnoid barrier cells, and other cell types did not reach statistical significance (*Figure 2—figure supplement 4*), the altered localization of CLDN5 and the spatial correlation between this mis-localization and BBB disruption (e.g. *Figure 5A*) suggests that increased plasma membrane protein internalization and/or degradation could play a role in EC barrier defects.

The NF-kappa B and WNT pathways can exhibit either positive and negative cross-regulation, depending on cell type and developmental context (*Ma and Hottiger, 2016*). By comparing BETA-CATENIN level, localization, and signaling in WT mouse embryo fibroblasts (MEFs) and in MEFs homozygous for inactivating mutations in IKKalpha or IKKbeta (the inflammation-activated kinases that phosphorylate the inhibitory binding partner of NF kappaB, leading to NF kappaB activation),

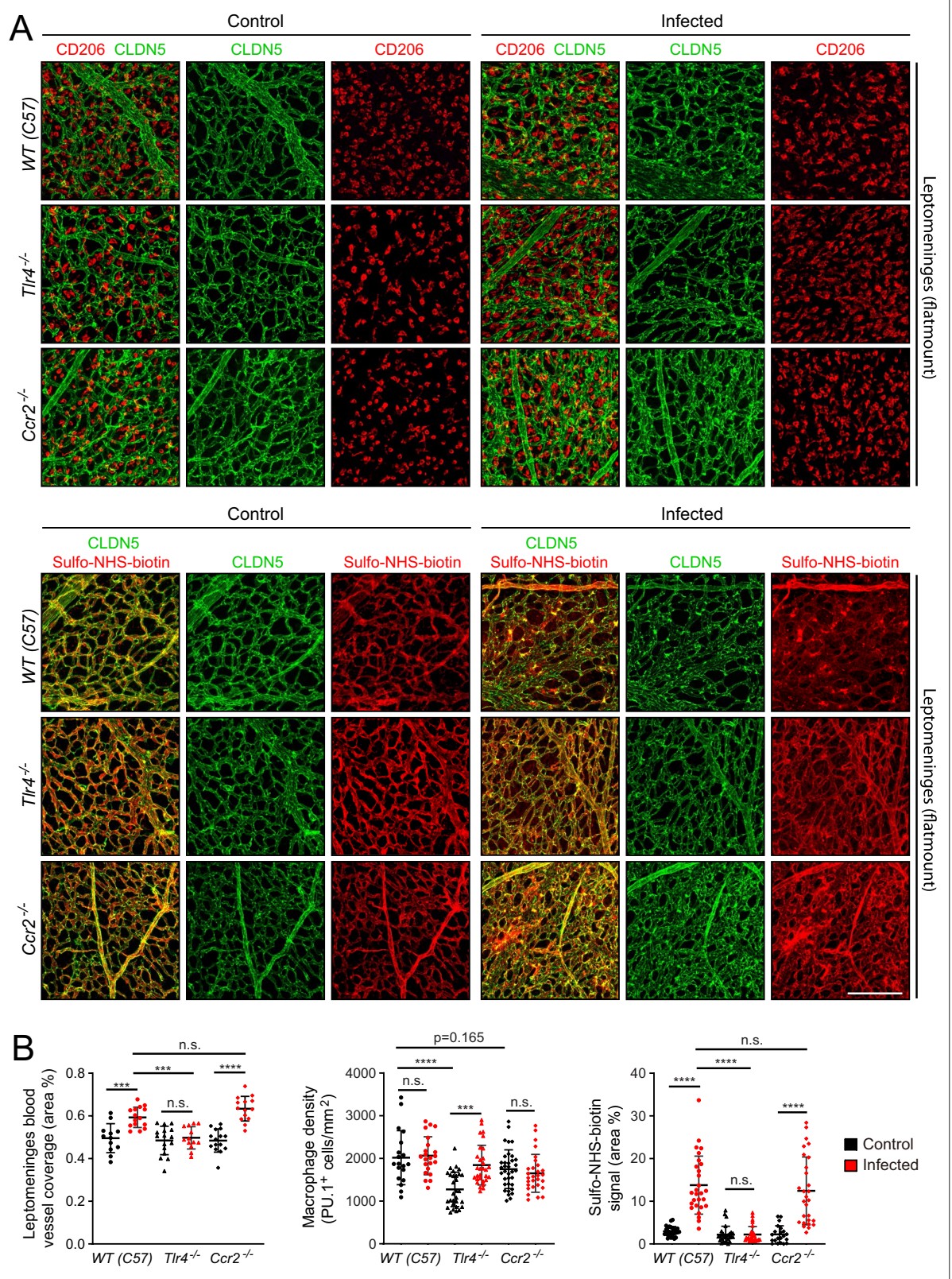

**Figure 7.** Effects of *Tlr4* KO and *Ccr2* KO on leptomeningeal EC responses to infection. (**A**) Leptomeninges flatmounts of control vs. infected mice showing, in the upper panels, macrophage density (CD206) and vascular architecture (CLDN5) and, in the lower panels, vascular leakage (sulfo-NHS biotin). (**B**) Quantification of (left) vascular architecture based on CLDN5 immunostaining, (center) macrophage density, and (right) Sulfo-NHS-biotin leakage in WT, *Tlr4⁻/⁻*, and *Ccr2⁻/⁻* leptomeninges flatmounts in control vs. infected mice. Scale bar: A, 100 μm.

*Figure 7 continued on next page*

*Figure 7 continued*

The online version of this article includes the following figure supplement(s) for figure 7:

**Figure supplement 1.** Expression of *Tlr4* and *Ccr2* in the major meningeal cell classes and in meningeal immune cells.

**Figure supplement 2.** Comparison of leptomeningeal macrophage quantification by counting PU.1+vs. CD206 + cells, using as a test case the experiments presented in *Figure 7*.

---

*Lamberti et al., 2001* found that IKKalpha and IKKbeta phosphorylate BETA-CATENIN on different sites and with opposite effects. IKKalpha phosphorylation leads to BETA-CATENIN stabilization and increased WNT signaling, whereas IKKbeta phosphorylation leads to BETA-CATENIN destabilization and decreased WNT signaling. These precedents in other cell types suggest that reduced LEF1 – and presumably reduced canonical WNT signaling – in ECs in the infected leptomeninges could reflect BETA-CATENIN down-regulation via the activated NF kappaB pathway. Reduced canonical WNT signaling in ECs would be predicted to reduce BBB integrity (*Rattner et al., 2022*).

The flat geometry of the meninges and its superficial location between the brain and the skull present unusually favorable opportunities for both in vitro and in vivo microscopy. For in vitro analyses, the relative thinness of the dissected mouse leptomeninges and dura provide excellent access to antibodies and also facilitate high quality confocal imaging without the need for chemical clearing agents, as described here. In vivo, thinned skull preparations that permit two-photon imaging of the mouse meninges have been described, and this approach can be applied to mice with fluorescent immune cells that have been inoculated with fluorescent bacteria to allow single-cell resolution in vivo imaging of bacterial meningitis in a native context (*Kjos et al., 2015*; *Coles et al., 2017b*; *Manglani and McGavern, 2018*). A largely unexplored opportunity also exists for ex vivo culture and live imaging of the dissected leptomeninges and dura (*Glimcher et al., 2008*). Such preparations could permit high-resolution imaging of (1) bacterial movement across the vascular wall, (2) interactions between bacteria and immune cells, and (3) interactions between host cells. The use of fluorescent reporters of signaling pathway activity would further increase the value of such analyses (*Kudo et al., 2018*; *Clark et al., 2021*).

Despite decades of research, numerous gaps remain in our understanding of the pathophysiology of bacterial meningitis. These include: (1) the ways in which the imature immune system differs from the more mature immune system in its response to infection, (2) the relative importance and the precise roles of different immune cells and immune modulators, and (3) the roles played by changes in gene expression and cell behavior among non-immune cell types. In each of these areas, mouse models of meningitis, together with new technologies for interrogating these models, can provide insights that inform the understanding of human meningitis.

## Materials and methods

### Mice

The following mouse lines were used: FVB/NJ (JAX#001800); C57BL/6 J (JAX#000664); Tie2-GFP (JAX#003658); B6(Cg)-Tlr4tm1.2Karp/J (JAX#029015); B6.129S4-Ccr2tm1Ifc/J (JAX#004999); and *Rosa26-NF-kappaB* reporter mice (described below). All mice were housed and handled according to the approved Institutional Animal Care and Use Committee protocol of the Johns Hopkins Medical Institutions. Meninges snRNA-seq experiments and histological studies used postnatal day 6 (P6) mice with age-matched controls.

### Construction and genotyping of the NF-kappa B reporter

To construct the Cre-dependent reporter for NF-kappa B signaling at the *Rosa26* locus, the following elements were inserted (from 5' to 3' in the order listed) into a standard *Rosa26* targeting vector: an *frt*-phosphoglycerate kinase (*Pgk*)-neomycin (*Neo*)-*frt* (*FNF*) cassette, which includes a strong poly-adenylation signal; five tandem repeats of a canonical NF kappaB response element (GGGACTTT CC); a minimal Delta Crystallin promoter followed by an intron; a *loxP-transcription stop-loxP* (*LSL*) cassette; an open reading frame coding for a nuclear-localization signal-tdTomato-3xHA protein (*nls-tdT-3xHA*); and a bovine growth hormone 3'UTR. The *Rosa26-NF-kappaB-LSL-tdT* targeting construct with a 3' flanking Diphteria toxin-A coding sequence was electroporated into R1 ES cells, which were

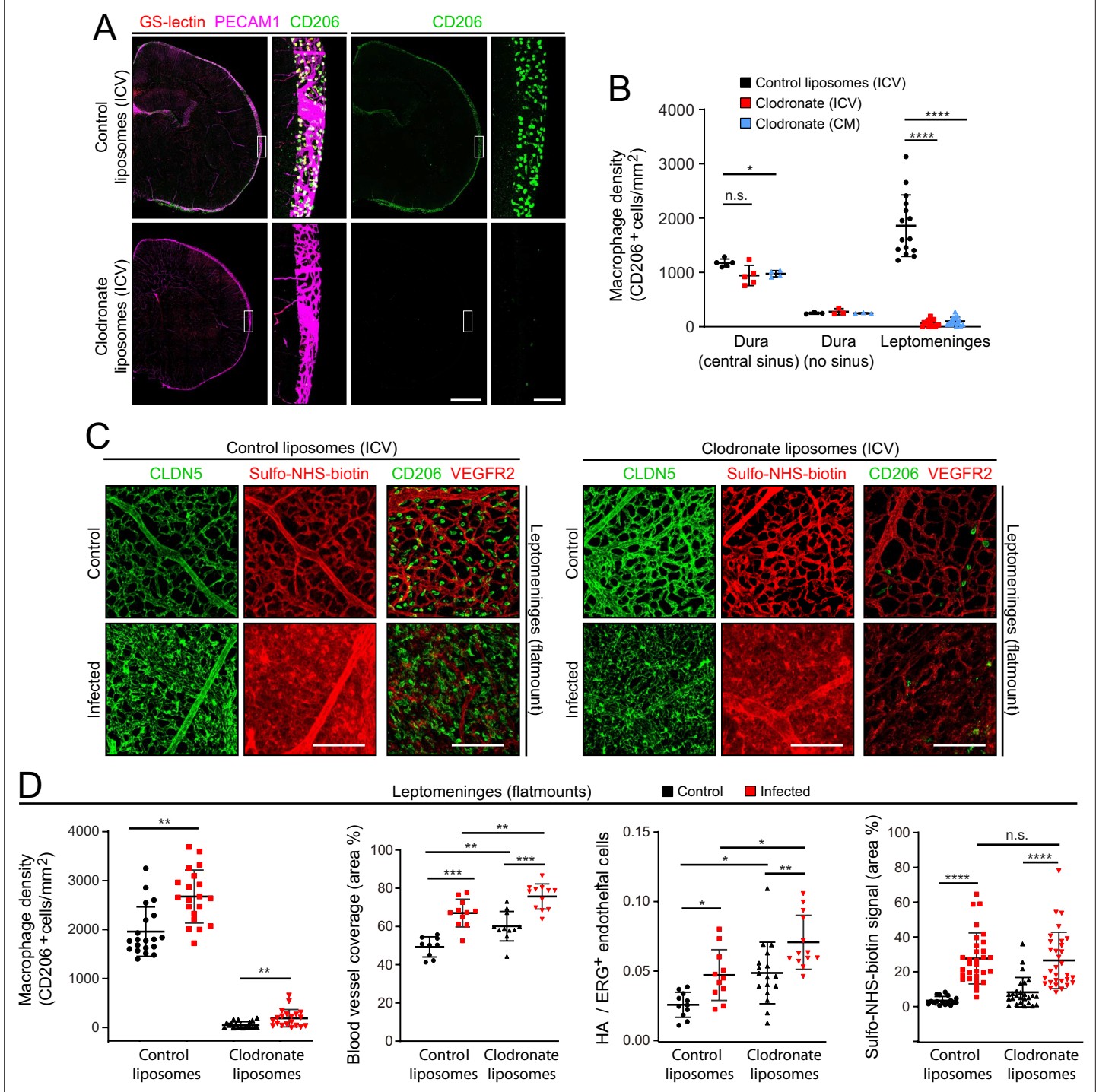

**Figure 8.** EC response to eliminating leptomeningeal macrophages with liposomal clodronate. (**A**) Intracerebroventricular (ICV) injection at P3 of empty liposomes vs. clodronate liposomes shows that clodronate almost completely eliminates leptomeningeal macrophages at P6, as visualized with CD206 immunostaining. (**B**) Quantification of leptomeningeal and dural macrophage abundance following ICV or cisterna magna (CM) injection of clodronate liposomes vs. control liposomes. (**C**) Leptomeninges flatmounts show greatly reduced numbers of macrophages in mice with or without infection following P3 treatment with clodronate liposomes, and there is little or no effect of macrophage depletion on CLDN5 relocalization and on Sulfo-NHS-biotin leakage in leptomeningeal ECs in response to infection. (**D**) Quantification in leptomeninges flatmounts from control vs. infected mice (from left to right): (1) macrophage density (2) vascular density, (3) NF-kappa B reporter activation in ECs (as shown in *Figure 6F*), and (4) Sulfo-NHS-biotin leakage. Mice received an ICV injection at P3 of empty liposomes or clodronate liposomes. CD206 immunostaining was used for macrophage quantification. *Figure 7—figure supplement 2* shows that counting PU.1 or CD206 immunostained cells gives closely similar results in leptomeninges flatmounts. Scale bars: A, 500 μm (low magnification) and 50 μm (inset); B, 100 μm.

then subjected to G418 selection. Clones harboring the targeted *Rosa26* locus were identified by Southern blot hybridization, karyotyped, and injected into blastocysts from Sv129 mice. Germline transmission to the progeny of founder males was determined by PCR. PCR primers for the parental allele: TGTCGGCCTGCAGCCAAAGCTTATCGA (sense, at the 3' end of the *Neo* casette) and TGAA GTTCTCAGGATCGGTCGCTA (antisense, in the intron). PCR primers for the Cre-recombined allele: CCCCTCTGCTAACCATGTTCATGCCTT (sense, in the intron) and GGCAACCTTCCTCTTCTTCTTAGG CATGGTGG (antisense, at the 5' end of the *nls-tdT-3xHA* open reading frame).

## Antibodies and other reagents

The following antibodies were used for tissue immunohistochemistry and immunoblotting: goat anti-CD45 (R&D Systems AF114-SP); rat anti-CD206/MRC1 (Bio-Rad MCA2235T); goat anti-CD206 (R&D Systems AP2535); rat anti-LYVE1 (Thermo Fisher/eBioscience 14-0443-82); rat anti-PU.1/Spi-1 (Novus Biologicals MAB7124); rat anti-CD180/RP105 antibody, PE (eBioscience 12-1801-81); goat anti-S100A8 (R&D Systems AF3059); rabbit anti-ERG (Cell Signaling Technologies 97249); rat anti-PECAM1/CD31 (BD Biosciences 553370); mouse anti-CLDN5, Alexa Fluor 488 conjugate (Invitrogen 352588); mouse anti-CLDN5 (Invitrogen 35–2500); rabbit anti-ZO1 (Invitrogen 40–2200); rabbit anti-LEF1 rabbit (Cell Signaling Technologies 2230); rabbit anti-LEF1, Alexa Fluor 647 conjugate (Cell Signaling Technologies 14022); rabbit anti-Occludin (Invitrogen 406100); goat anti-VEGFR2/KDR (R&D Systems AF644-SP); sheep anti-FOXP2 (R&D Systems AF5647-SP); rabbit anti-SATB2 (Abcam ab92446); rat anti-IL-6 (Biolegend 504501); rabbit anti-COL14A1 (Novus Biologicals NBP2-15940); chicken anti-GFP (Abcam Ab13970), rat anti-HA (Proteintech 7c9); rabbit anti-HA (homemade); goat anti-E-cadherin (R&D Systems AF748); rabbit anti-E-cadherin (Cell Signaling Technologies 3195); rabbit anti-AIFM3 (Novus Biologicals NBP1-76889); rabbit anti-COL25A1 (G-Biosciences ITT1021); goat anti-IGSF8 (R&D systems AF3117-SP); rabbit anti-NNAT (Abcam ab27266); mouse anti-GAPDH (Cell Signaling Technologies 97166 S); rabbit anti-GAPDH (Cell Signaling Technologies 5174 S); Streptavidin, Alexa Fluor 488 conjugate (Invitrogen S11223); Streptavidin, Alexa Fluor 647 conjugate (Invitrogen S32357). Alexa-Fluor-conjugated secondary antibodies were from Invitrogen. Infrared immunoblotting secondary antibodies were from LI-COR.

Other reagents used: Sulfo-NHS-biotin (Thermo Fisher Scientific #21217); LPS O111:B4 (Sigma-Aldrich L2630); Benzonase Nuclease, ultrapure (Sigma-Aldrich, E8263-5KU); clodronate and control liposomes: Mannosylated Macrophage Depletion kit (Encapsula NanoScience SKU#CLD-8914); Micro BCA Protein Assay kit (ThermoFisher Scientific 23235).

## *E. coli* infection and LPS injection

*E. coli* strain RFP-RS218 (O18:K1:H7) with K1 capsule is a clinical isolate from the CSF of a neonate with meningitis (*Zhu et al., 2020*; a generous gift from the late Dr. Kwang Sik Kim, Johns Hopkins Medical School, Baltimore, MD). *E. coli* were grown overnight in Luria broth containing 100 µg/ml ampicillin at 37 °C. The following day, 400 µL of the *E. coli* culture was added to 20 mL of fresh Luria broth for an additional 2 hr of culture at 37 °C. The bacteria were washed one time in PBS, the OD at 620 nm was measured, and the concentration of the samples was adjusted prior to inoculation.

For most experiments, *E. coli* meningitis was induced in FVB/NJ mice. For experiments with *Tlr4*[-/-] and *Ccr2*[-/-] mice, C57BL/6 J was used as the control to match the strain background of the KO lines. Briefly, a litter of P5 mice were randomly divided into two groups that were either subcutaneously injected in the back with $1.2 \times 10^5$ CFU of *E. coli* RFP-RS218 in 20 µL PBS or not injected. Twenty-two hours later, the mice were sacrificed as described. *Tlr4*[-/-] and *Ccr2*[-/-] on a C57BL/6 J background and *Rosa26-NF-kappaB* reporter mice on a mixed Sv129 x C57BL/6 J background were subcutaneously injected with $8 \times 10^4$ CFU of *E. coli* RFP-RS218. For LPS-injection, P5 mice were intraperitoneally injected with a single dose of LPS O111:B4 (10 mg/kg) or, for control mice, the same volume of PBS. Twenty-two hours later (at P6), the mice were sacrificed as described. All analyses (snRNAseq and histology) were conducted on mice that were sacrificed at P6.

## Meningeal macrophage depletion with clodronate liposomes

CNS macrophages were depleted as described in *Polfliet et al., 2001*. Each mouse was injected with control or clodronate liposomes (3 µl of a 5 mg/ml stock solution) two days before being infected with *E. coli* or before receiving an LPS injection. Liposomes were allowed to warm from refrigerator

temperature to room temperature for 1 hr prior to injection. P3 mice were deeply anesthetized on ice and then slowly injected with liposomes using a Hamilton syringe (Hamilton Bonaduz, AG Switzerland) either in a lateral ventricle (intracerebroventricular administration) or in the cisterna magna.

## Tissue processing

To prepare isolated dura and leptomeninges tissues for snRNAseq, immunoblots, and whole mount immunostaining, mice were deeply anesthetized on ice and then perfused via the cardiac route with PBS. The skullcaps (with dura attached) and brain (with leptomeninges attached) were dissected in PBS. The brain was then chilled in ice-cold PBS for several minutes. Using a fine tweezers, the leptomeninges was gently peeled from the surface of the brain and either used for protein extraction, or immersion fixed in 2% paraformaldehyde (PFA)/PBS at room temperature for 1 hr for subsequent immunostaining. The dura was gently peeled from the skullcap and immersion fixed in 2% paraformaldehyde (PFA)/PBS at room temperature for 1 hr for subsequent immunostaining. Alternately, the skullcap and attached dura were immersion fixed overnight in 2% PFA/PBS at 4 °C without further dissection. Following immersion fixation, each sample was washed three times in PBS. For vibratome sections, the brain and attached leptomeninges was fixed overnight in 2% PFA/PBS at 4 °C, washed the following day in PBS at 4 °C for at least 3 hr, embedded in 3% agarose, and sectioned at 150 µm thickness using a vibratome (Leica).

For analysis of vascular leakage, P6 mice were injected intraperitoneally with Sulfo-NHS-biotin (30 µl of 20 mg/ml Sulfo-NHS-biotin in PBS per mouse) 10–15 min before sacrifice. After IP injection, the tracer rapidly equilibrates into the systemic circulation. Mice were deeply anesthetized on ice and perfused via the cardiac route with PBS followed by leptomeninges dissection and PFA fixation, as described above.

For cross-sections of isolated dura and leptomeninges, fixed tissues were embedded in optimal cutting temperature compound (OCT, Tissue-Tek), rapidly frozen in dry ice, and stored at –80 °C. Thirty µm sections were cut on a cryostat and thaw-mounted onto Superfrost plus slides. Slides were stored at –80 °C until further processing.

## Immunohistochemistry

Whole mount leptomeninges, dura, or brain sections were incubated overnight at 4 °C with primary antibodies diluted with PBSTC (PBS with 1% Triton X-100, 0.1 mM CaCl$_2$) plus 10% normal goat serum (NGS). Tissues were washed four times with PBSTC over the course of 6–8 hours, and then incubated overnight at 4 °C with secondary antibodies diluted in PBSTC plus 10% NGS. If a primary rat antibody was used, secondary antibodies were additionally incubated with 0.5% normal mouse serum as a blocking agent. The following day, tissues were washed at least four times with PBSTC over the course of 6–8 hr, flat-mounted on Superfrost Plus glass slides (Fisher Scientific), and coverslipped with Fluoromount G (EM Sciences 17984–25).

For leptomeninges or dura cross-sections, sections on slides were covered with 2% PFA/PBS at room temperature for 15 min, washed three times in PBS, and incubated overnight with primary antibodies diluted in PBSTC plus 10% NGS at 4 °C. The following day, sections were washed at least four times with PBSTC and incubated with secondary antibodies for 2 hr at room temperature. Sections were then washed four times with PBSTC and coverslipped with Fluoromount G. For each immunostaining analysis, whole mounts and sections were stained from at least two independent experiments.

## Confocal microscopy

Confocal images were captured with a Zeiss LSM700 confocal microscope (20x and 63x objectives) using Zen Black 2012 software, and processed with Fiji-ImageJ, Adobe Photoshop, and Adobe Illustrator. The depths of the Z-stacked flatmount images from the confocal series were chosen to capture the full thickness of the tissue: (1) 40–60 µm for the dura sinus region, (2) 15–20 µm for the dura peripheral region, and (3) 20–30 µm leptomeninges. Each point in the quantification of immunofluorescent data represents the analysis of a single Z-stacked confocal image from a flatmount that encompasses the full depth of the tissue (leptomeninges or dura), unless noted otherwise. For Sulfo-NHS-biotin detection with streptavidin, ~twofold animal-to-animal variability is typically seen in the overall intensity of the strepatvidin signal, most likely due to variable uptake of Sulfo-NHS-biotin from the site of IP

injection. To permit a clearer comparison between images of Sulfo-NHS-biotin leakage, this intensity variation has been minimized by manually adjusting the brightness of the Sulfo-NHS-biotin channel.

## Leptomeninges tissue lysate

To prepare leptomeninges proteins for western blotting, anesthetized mice were perfused with PBS and their brains were dissected in PBS. The leptomeninges was detached from the surface of the brain tissue with tweezers and then transferred into a 1.5 mL eppendorf tube and stored at –80 °C for further processing. Frozen leptomeninges from two mice were pooled, lysed in lysate buffer (50 mM Tris-HCl (pH, 7.4), 150 mM NaCl, 2 mM $MgCl_2$, 1% Triton X-100, 0.25 U/µL Benzonase), and then homogenized using a plastic pestle fitted for eppendorf tubes. The homogenates were incubated for 10 min at room temperature to digest the nuclear DNA, and then SDS was added to a final concentration of 0.5%. The lysate in SDS was incubated for 20 min at 4 °C and then centrifuged at 14,000 x $g$ for 15 min at 4 °C. The supernatant was recovered and its protein concentration was determined using the BCA protein assay kit.

## Immunoblotting

Protein samples (6–8 µg per sample) were loaded onto a 4–12% NuPAGE Bis-Tris protein gel, which was run at 130 V for 1.5 h and then blotted onto a nitrocellulose membrane (Millipore). Membranes were blocked with Intercept blocking buffer (LI-COR 927–60001) at room temperature for 1 hr and then probed overnight with primary antibodies diluted in blocking buffer at 4 °C. The following day, the membranes were washed four times with TBST and then probed at room temperature for 2 h with the corresponding infrared secondary antibodies diluted in blocking buffer. Bands were visualized with an Odyssey Fc Imager (LI-COR) and band intensities were quantified with Fiji-Image J software.

## snRNAseq

Two and three independent biological replicate libraries were prepared for the control and *E. coli*-infection groups, respectively, with one P6 mouse used per library. For each sample, the dura and leptomeninges were rapidly dissected in ice-cold DPBS (Gibco 14287072). The combined dura and leptomeninges were minced with a razor blade and Dounce homogenized using a loose-fitting pestle in 5 mL homogenization buffer (0.25 M sucrose, 25 mM KCl, 5 mM MgCl2, 20 mM Tricine-KOH,pH 7.8) supplemented with 1 mM DTT, 0.15 mM spermine, 0.5 mM spermidine, EDTA-free protease inhibitor (Roche 11836 170 001), and 60 U/mL RNasin-Plus RNase Inhibitor (Promega N2611). A 5% IGEPAL-630 solution was added to bring the homogenate to 0.3% IGEPAL CA-630, and the sample was further homogenized with ten strokes of a tight-fitting pestle. The sample was filtered through a 50 µm filter (CellTrix, Sysmex, 04-004-2327), underlayed with solutions of 30% and 40% iodixanol (Sigma D1556) in homogenization buffer, and centrifuged at 10,000×$g$ for 18 min in a swinging bucket centrifuge at 4 °C. Nuclei were collected at the 30–40% interface, diluted with two volumes of homogenization buffer, and concentrated by centrifugation for 10 min at 500x$g$ at 4 °C. snRNAseq libraries were constructed using the 10 x Genomics Chromium single-cell 3' v3 kit following the manufacturer's protocol (https:// support. 10xgenomics. com/ single- cell- gene- expression/ library- prep/ doc/ user- guide- chromium-single- cell- 3- reagent- kits- user- guide- v31- chemistry). Libraries were sequenced on an Illumina NovaSeq 6000.

## Analysis of snRNAseq data

Reads were aligned to the mm10 pre-mRNA index using the Cell Ranger count program, version 3.1.0. The data for the different libraries was merged using the Cell Ranger merge command. Data analysis was performed using the Seurat R package, version 4.0.1 in RStudio. After filtering out nuclei with >1% mitochondrial transcripts or with <500 or>6000 genes, 48,941 nuclei were retained, 14,357 nuclei from the two control samples and 34,585 nuclei from the three infected samples. The data were normalized using a regularized negative binomial regression algorithm implemented in the SCTransform function as described in *Hafemeister and Satija, 2019*. UMAP dimensional reduction was performed using the R uwot package (https://github.com/jlmelville/uwot) integrated into the Seurat R package (*Melville, 2022*). To compare cell types across treatments, the data was integrated using the strategy described in *Stuart and Satija, 2019*. This pipeline involves splitting the dataset by treatment using the Seurat SplitObject function and integrating the subset

objects using FindIntegrationAnchors and IntegrateData functions. Data for the various scatter plots were extracted using the Seurat AverageExpression function, and differential gene expression was analyzed using the Seurat FindMarkers function. The Wilcoxon Rank Sum test was used to calculate p-values. The p-values were adjusted with a Bonferroni correction using all genes in the dataset. Data exploration, analysis, and plotting were performed using RStudio (RStudio Team, 2020), the tidyverse collection of R packages (*Wickham, 2017*), and ggplot2 (*Wickham, 2009*). Dotplots were generated with the default settings, including default normalization. For Gene set enrichment analysis (GSEA; *Subramanian et al., 2005*), genes were ranked by the fold expression change between control and infected datasets. The ranked gene list was used to detect enriches gene sets within the Broad Institute Hallmark Gene Sets using the fgsea R package (https://github.com/ctlab/fgsea; *Korotkevich et al., 2019*).

## Macrophage quantification in the meninges

The density of macrophages was quantified using Fiji-ImageJ (https:// imagej. net/ software/ fiji/) from captured Z-stacked leptomeninges or dura wholemounts. The numbers of PU.1+or CD206+ cells were quantified in representative regions and then normalized to the area analyzed.

## Cell orientation analysis

Using Fiji-ImageJ, the axis of blood flow of the midline vein located in the superior sagittal sinus was defined as 0°, and then the long axis of each EC nucleus, identified by ERG immunostaining, was scored and its angle calculated relative to 0°. Nuclei were analyzed from three independent control vs. infection experiments.

## Fraction of area covered by blood vessels

The relative area covered by leptomeningeal blood vessels was determined from Z-stacked flatmount images that had been immunostained with CLDN5 or PECAM1. More specifically, flatmount images of representative regions that were populated by capillaries, but not by large veins or arteries, were overlayed in Adobe Illustrator with arrays of 10 parallel and evenly-spaced straight white lines. The length of each line corresponds to 166 µm on the image and the first and tenth lines in each array are separated by a distance that corresponds to 150 µm on the image. Thus, the 10 lines define a 166 µm x 150 µm rectangle. Along each white line, the widths of all of the regions in the image that were not covered by blood vessels were manually scored by drawing (in Adobe Illustrator) a straight line across the vessel-free region. When all of the vessel-free line segments had been drawn for a given square array, the sum of their lengths was calculated with Fiji-ImageJ and divided by the sum of the lengths of the ten white lines to generate an estimate for the fraction of the area not covered by blood vessels. The fraction of the area covered by blood vessels equals one minus the fraction of the area not covered by blood vessels. Each data point in a blood vessel area plot represents the area estimate from one set of white lines, that is the estimate obtained from sampling a length of 10x166 µm=1.66 mm.

## Statistical analysis

All statistical values are presented as mean ± SD. The number of mice used for each sample are listed in *Supplementary file 4*. The Wilcoxon rank sum test was used to measure statistical significance, except for *Figure 5D and G*, in which the sample size is too small and the student t-test was used instead. Statistical tests were carried out using the following web sites: https://www.socscistatistics.com/tests/signedranks/default2.aspx and https://www.omnicalculator.com/statistics/wilcoxon-rank-sum-test#how-do-i-calculate-wilcoxon-rank-sum-test. The statistical significance is represented graphically as n.s., not significant (i.e. $p > 0.05$); *, $p < 0.05$; **, $p < 0.01$; ***, $p < 0.001$; ****, $p < 0.0001$.

## Acknowledgements

Supported by the Howard Hughes Medical Institute. The authors thank Dr. Latika Nagpal for helpful comments on the manuscript. The authors thank the reviewers and the editors for excellent comments that improved the manuscript.

## Additional information

### Funding

| Funder | Grant reference number | Author |
|---|---|---|
| Howard Hughes Medical Institute | | Jeremy Nathans<br>Jie Wang<br>Amir Rattner |
| National Eye Institute | R01EY018637 | Jeremy Nathans |

The funders had no role in study design, data collection and interpretation, or the decision to submit the work for publication.

### Author contributions

Jie Wang, Conceptualization, Formal analysis, Investigation, Methodology, Writing - original draft, Writing - review and editing; Amir Rattner, Formal analysis, Investigation, Writing - review and editing; Jeremy Nathans, Conceptualization, Formal analysis, Supervision, Funding acquisition, Writing - original draft, Project administration, Writing - review and editing

### Author ORCIDs

Amir Rattner ⓘ http://orcid.org/0000-0001-9542-6212
Jeremy Nathans ⓘ http://orcid.org/0000-0001-8106-5460

### Ethics

All mice were housed and handled strictly according to the approved Institutional Animal Care and Use Committee protocol of the Johns Hopkins Medical Institutions (Protocol MO19M429).

### Decision letter and Author response

Decision letter https://doi.org/10.7554/eLife.86130.sa1
Author response https://doi.org/10.7554/eLife.86130.sa2

## Additional files

### Supplementary files

• Supplementary file 1. snRNAseq library statistics.

• Supplementary file 2. Criteria for assigning cell type clusters based on snRNAseq transcript profiles.

• Supplementary file 3. Differential transcript abundances among the major meningeal cell types in infected vs. control snRNAseq datasets. FC, fold change.

• Supplementary file 4. Number of mice in each group in the image quantifications.

• Supplementary file 5. Differential transcript abundances among CCL2$^-$ and CCL2$^+$ resident macrophages in infected vs. control snRNAseq datasets. FC, fold change.

• Supplementary file 6. Differential transcript abundances among EC cell types in infected vs. control snRNAseq datasets. FC, fold change.

• MDAR checklist

### Data availability

Sequencing data have been deposited in GEO.

The following datasets were generated:

| Author(s) | Year | Dataset title | Dataset URL | Database and Identifier |
|---|---|---|---|---|
| Wang J, Rattner A, Nathans J | 2022 | Bacterial meningitis in the early postnatal mouse studied at single-cell resolution | https://www.ncbi.nlm.nih.gov/geo/query/acc.cgi?acc=GSE221678 | NCBI Gene Expression Omnibus, GSE221678 |

*Continued on next page*

*Continued*

| Author(s) | Year | Dataset title | Dataset URL | Database and Identifier |
|---|---|---|---|---|
| Wang J, Rattner A, Nathans J | 2022 | snRNAseq_JW19_meninges_control_RP1 | https://www.ncbi.nlm.nih.gov/geo/query/acc.cgi?acc=GSM6892910 | NCBI Gene Expression Omnibus, GSM6892910 |
| Wang J, Rattner A, Nathans J | 2022 | snRNAseq_JW20_meninges_control_RP2 | https://www.ncbi.nlm.nih.gov/geo/query/acc.cgi?acc=GSM6892911 | NCBI Gene Expression Omnibus, GSM6892911 |
| Wang J, Rattner A, Nathans J | 2022 | snRNAseq_JW21_meninges_infected_RP1 | https://www.ncbi.nlm.nih.gov/geo/query/acc.cgi?acc=GSM6892912 | NCBI Gene Expression Omnibus, GSM6892912 |
| Wang J, Rattner A, Nathans J | 2022 | snRNAseq_JW22_meninges_infected_RP2 | https://www.ncbi.nlm.nih.gov/geo/query/acc.cgi?acc=GSM6892913 | NCBI Gene Expression Omnibus, GSM6892913 |
| Wang J, Rattner A, Nathans J | 2022 | snRNAseq_JW23_meninges_infected_RP3 | https://www.ncbi.nlm.nih.gov/geo/query/acc.cgi?acc=GSM6892914 | NCBI Gene Expression Omnibus, GSM6892914 |

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
