## [Editor Report]

This study presents valuable findings on the changes in immune cell populations and stromal cells occurring at the CNS borders in a neonatal bacterial meningitis model, focusing on fibroblasts, macrophages, and endothelial cells. The study provides a solid snRNA-seq dataset and high-quality immune fluorescence images of dissected brain border regions, that will be useful for the community. These observations and datasets will be of interest to the neuro-immunology community.

---

## [Decision Letter]

**Decision letter after peer review:**

Thank you for submitting your article "Bacterial meningitis in the early postnatal mouse studied at single-cell resolution" for consideration by *eLife*. Your article has been reviewed by 2 peer reviewers, and the evaluation has been overseen by a Reviewing Editor and Marianne Bronner as the Senior Editor. The reviewers have opted to remain anonymous.

Essential revisions:

1. Resident Ccl2+ and Ccl2- macrophages should be better characterised/described in

infected state and uninfected state.

2. A clearer definition of arachnoid as reviewer think that the authors investigated arachnoid+pia, which represent leptomeninges.

3. Comparative analysis of dural versus leptomeningeal response: To measure the bacterial load and its dissimination in dura+arachnoid+pia+parenchyma in WT, TLR4-/-, CCR2KO and clodronate-treated mice (d1, 4, 10 if possible).

4. Immunofluorescence images should be quantified throughout the manuscript.

5. Time points of analysis, repetitions and number of mice used should be clearly stated in the figure legends.

*Reviewer #1 (Recommendations for the authors):*

1. The definition of the arachnoid is the manuscript is unclear. It seems that the authors studied and peeled the leptomeninges (pia+arachnoid) so it is possible that the manuscript is mainly focusing on the pia or leptomeninges, not the arachnoid. There also seems to be some controversy in the literature as to, when removing the skull cap, the arachnoid remains attached to the pia of the brain and/or to the dura of the skull. This should be better explained and justified.

2. In the RNAseq data, how can the authors know that the clusters of cells (e.g. macrophages) define cell types versus cell states? Without lineage tracing, it seems impossible to clearly define distinct population of macrophages and to follow their trajectory and transcriptomic changes. It is likely that after activation, some macrophages of population A fall into another cluster of population B, thus biasing the analysis of cluster B transcriptomic changes. Thus the quantification of populations and DEG based should be avoided when the population' clusters are not clearly distinct. This applies for Fibro dura (Figure 1), Macrophage (Figure 3A) and Ec dura (Figure 4A). Only the whole population/cluster can be analyzed.

3. It would be interesting to have a bigger picture about each of the 3 cell types of interest (macrophages, endothelium, fibroblasts) by comparing leptomeningeal versus dural response. For each cell type, how many DEG are shared between leptomeningeal versus dural tissue? How many are unique to each cell type/cell location?

4. The authors should quantify by histo or RTqPCR the bacterial load in the dural versus leptomeninges.

5. All this is 1 day post-infection. There should be an important monocyte and neutrophil infiltration in the dura and the leptomeninges. Does the data show it? Could infiltrates contribute to the transcriptomic changes through cytokine release?

6. Did collagen stain decrease upon infection, as suggested by RNAseq? The authors should also validate their markers for endothelial subtypes by IF. Also, it's unclear how veins and arteries were distinguished, and SMA stain should be used to confirm.

7. What was the consequence of TLR4 KO, CCR2 KO and clodronate on bacterial load? Maybe 1 day post infection is too early.

8. The discussion would need to incorporate all the data and show the big picture: are meningeal layers similarly 1/infected, 2/responsive, 3/leaky? What could be the consequence of the transcriptomic changes (collagen, slc transporters, etc.)?

9. What happens at later time points? Which barrier breaks the most and lets bacteria invade the tissue?

*Reviewer #2 (Recommendations for the authors):*

– It is not clear what the 'Ccl2+ and Ccl2-' resident macrophages are. Do they correlate to the MHCII+ and MHCII- dural macrophages? Which other markers that were previously identified by these subsets in adult mice, do they express? Are they present in both the dura and the arachnoid/pia mater? In the dura, do they localize to the sinus region? How does this compare to adult mice?

– They mention that IL-6 was expressed by fibroblasts. However, in the image provided in Figure 2H, it cannot be concluded which cells express IL-6, apart from being negative for CD206.

– In Figure 3, it is difficult to see the changes across immune cells in infected vs. control. Why not show a pie chart, or also UMAPs of infected vs. control side by side, or overlayed with a color (as in Figure 4A).

– For snRNA-seq, n=? how many mice were used?

– Figure 4, does this include both meningeal compartments together?

– it is difficult to quantitively assess the frequencies of the cell populations from the sequencing data. To validate the dynamics of different cell types, validation using flow cytometry (total cell counts) would be nice to include.

– The authors explain the increase of macrophages in infection due to conversion of resident macrophages to inflammatory macrophages. Fate-mapping resident macrophages or monocytes prior to infection would be helpful to substantiate this claim.

– Figure 7, the difference of CLDN5 between Ccr2-/- control and infected is not evident in the images (A), yet the quantification shows a highly significant difference (B). This should be explained. In the images in (A), a WT control for both infected and non-infected should be shown.

– Figure 7C, Tlr4-/- mice have fewer meningeal macrophages already at steady state. Can they speculate why and discuss this?

– Also, in Ccr2-/- mice, fewer macrophages (CD206+ or PU.1+ cells) were found in the meninges, of both infected and non-infected mice. Is there a difference between the subsets they identified?

– In Tlr4-/- and Ccr2-/- or clodronate-treated mice, bacteria dissemination in the meninges and brain should also be analyzed.

– They suggest that Tlr4 is important on non-macrophages. Can they show in their snRNA-seq data which cell types express it?

– It should be discussed that killing of macrophages (by clodronate) treatment, may also lead to inflammation and changes in ECs (as seen in Figure 8E).

– Immunofluorescence images should be quantified throughout the manuscript. (For example, this is lacking in Figure 2F,G, Figure 5E,F).

– Figure 5G, for LEF, n=2, this would need to be increased to show statistical significance

– Time point of analysis, repetitions and number of mice used is not always clear in the figure legends.

---

## [Author Response]

Essential revisions:1. Resident Ccl2+ and Ccl2- macrophages should be better characterised/described ininfected state and uninfected state.

We have expanded the description of these cells in the Results section and added a new

Supplemental Figure 8 to illustrate the behavior of transcripts relevant to these macrophage subtypes.

2. A clearer definition of arachnoid as reviewer think that the authors investigated arachnoid+pia, which represent leptomeninges.

Thank you for pointing this out. The reviewer is correct – what we have dissected and studied is the leptomeninges (arachnoid + pia), not just arachnoid. This has now been changed throughout the text and figures.

3. Comparative analysis of dural versus leptomeningeal response: To measure the bacterial load and its dissimination in dura+arachnoid+pia+parenchyma in WT, TLR4-/-, CCR2KO and clodronate-treated mice (d1, 4, 10 if possible).

This has been now been done by counting the number of bacteria in tissue – since the bacteria express RFP, they are readily seen in the confocal images. These data are presented in Supplemental Figure 3.

4. Immunofluorescence images should be quantified throughout the manuscript.

Thank you for this point. We agree. This has now been done, with additional panels added to the Figures 2, 5, 7, and 8.

5. Time points of analysis, repetitions and number of mice used should be clearly stated in the figure legends.

A good point. This has now been done. Putting this information in the figure legends is going to clutter the legends, so we have put together a new Supplementary Table (Supplementary Table 4) that has this information for the entire manuscript. We also define more precisely in the legend to Figure 2 and in the Methods section exactly what is being represented by each data point: “Each point in this and other quantifications of immunofluorescent data represents the analysis of a single Z-stacked confocal image that encompasses the full depth of the tissue (leptomeninges or dura), unless noted otherwise.” The time points are all P6. This is now clearly stated in the Results section.

Reviewer #1 (Recommendations for the authors):1. The definition of the arachnoid is the manuscript is unclear. It seems that the authors studied and peeled the leptomeninges (pia+arachnoid) so it is possible that the manuscript is mainly focusing on the pia or leptomeninges, not the arachnoid. There also seems to be some controversy in the literature as to, when removing the skull cap, the arachnoid remains attached to the pia of the brain and/or to the dura of the skull. This should be better explained and justified.

Thank you for this comment. The reviewer is correct – it is leptomeninges, not arachnoid.

This has now been changed throughout the text and figures. Regarding the dissection, at the start of the Results section, the text addresses this issue. When the skull cap is removed, the arachnoid and pia (=leptomeninges) remains with the brain and the dura remains with the skull – i.e. the tissue splits between the arachnoid and the dura. Figure 1B-E demonstrates this. At the start of the Results section, the text now states: “When the skull and brain are separated in the absence of fixation, the natural cleavage plane is between the leptomeninges and dura (Figure 1A). The leptomeninges can then be peeled from the brain surface and the dura can be peeled from the inner surface of the skull.”

2. In the RNAseq data, how can the authors know that the clusters of cells (e.g. macrophages) define cell types versus cell states? Without lineage tracing, it seems impossible to clearly define distinct population of macrophages and to follow their trajectory and transcriptomic changes. It is likely that after activation, some macrophages of population A fall into another cluster of population B, thus biasing the analysis of cluster B transcriptomic changes. Thus the quantification of populations and DEG based should be avoided when the population' clusters are not clearly distinct. This applies for Fibro dura (Figure 1), Macrophage (Figure 3A) and Ec dura (Figure 4A). Only the whole population/cluster can be analyzed.

This is a general issue with any cluster analysis. Here are two general comments as a prelude to the more specific response in the next paragraph. First, it is important to remember that the UMAP plots are dimensionally reduced representations that allow high dimensional data sets to be approximated and visualized in a 2D plot, and that these plots under-represent the distinctions between clusters in the same way that principal component analysis under-represents distinctions when one looks only at the first two components. The real data, which is the basis for the clustering, is a very large matrix that cannot be visualized. Second, the reviewer correctly notes that we have not performed a lineage tracing analysis to determine the dynamics of cell movements. Our data is silent on that issue. This limitation is now noted in both the Results and the Discussion sections.

The clusters in Figures 1F, 3A, and 4A are generated by Seurat, using default parameters – the result is that some clusters are obvious and others are less obvious when visualized in a dimensionally reduced form with a UMAP plot. The distinction between clusters is supported by the specificity of expression of individual transcripts, as illustrated in Figures 1I, 3B, 4C, and Supplemental Figure 8A. To address the reviewer’s comment with respect to macrophages, the distinction between the four macrophage subtypes within the large macrophage cluster in the UMAP plot in Figure 3A is not visually apparent – they all seem to be part of a single large cluster when the N-dimensional space is reduced to two dimensions. However, the dot plot in Figure 3B shows that the subdivisions delineated by Seurat are based on objective differences in gene expression. Additionally, the UMAP plots in Figure 3F show a compact crescent-shaped macrophage cluster in the control samples that changes to a broad disc shaped cluster in the infected state, indicative of substantial changes in gene expression. Our conclusion is that these changes in macrophage gene expression are real. That said, we agree with the reviewer’s general point that the macrophage clustering is relatively subtle, which is not entirely surprising, since these clusters represent subtypes or, perhaps more accurately, gene expression substates of macrophages. We have added the following sentence to the Results section: “These data are consistent with a model in which infection promotes the appearance of new macrophages “states”, as defined by novel patterns of gene expression that are distinct from those of resting macrophages.”

In the final analysis, clustering is simply a heuristic tool, a way of organizing the data so that we can more easily think about it. We note that this issue is one that neurobiologists have been wrestling with for decades in their analyses of neuronal classes (an example is below).

Brain Behav Evol 1983;23(3-4):121-64.

R W Rodieck, R K Brening

Retinal ganglion cells: properties, types, genera, pathways and trans-species comparisons Abstract.

This article attempts to develop an empirical foundation for the notion of 'cell type', and to use this notion to clarify the issues involved in the classification, pathways, and trans-species comparisons of retinal ganglion cells.

3. It would be interesting to have a bigger picture about each of the 3 cell types of interest (macrophages, endothelium, fibroblasts) by comparing leptomeningeal versus dural response. For each cell type, how many DEG are shared between leptomeningeal versus dural tissue? How many are unique to each cell type/cell location?

Thank you for this suggestion. We have now done this in Supplemental Figure 5, which shows DEG genes (in this case, any gene with a log2-fold change greater than or equal to 2.5 in any of the listed cell types) for leptomeninges and dura endothelial cells (ECs), for leptomeninges and dura fibroblasts, and for arachnoid barrier cells. There are a core of DEGs with shared responses, as well as DEG differences that distinguish ECs from fibroblasts. A more comprehensive visualization of DEGs is shown in scatterplots in Supplemental Figure 4, and the actual data is tabulated in Supplemental Tables 3, 5, and 6. We have not separated macrophages based on location (dura vs. leptomeninges) because we did not perform the snRNAseq with these two tissues separately, and there are no published markers for macrophages at P6 that distinguish macrophages between these tissues. We have not directly answered the question “how many DEGs are shared” because the answer depends on the (arbitrary) cut-off for the fold-difference that constitutes a DEG. With a cut-off of log2-fold change >2.5, Supplemental Figure 5 gives a reasonable sense of the relatedness of the responses. This is now described in the Results section.

4. The authors should quantify by histo or RTqPCR the bacterial load in the dural versus leptomeninges.

This has been now been done by counting the number of bacteria in tissue – since the bacteria express RFP, they are readily seen in the confocal images. These data are presented in Supplemental Figure 3.

5. All this is 1 day post-infection. There should be an important monocyte and neutrophil infiltration in the dura and the leptomeninges. Does the data show it? Could infiltrates contribute to the transcriptomic changes through cytokine release?

In this study, we have not addressed the movement of immune cells between compartments. We agree that some cells in both the leptomeninges and the dura could be recent arrivals, but our data present only a static image, and without a cell tracing analysis one cannot assess the dynamics of cell movement. We note that a cell tracing analysis in the meninges is technically non-trivial as there is good evidence that the pool of myeloid cells that supplies the meninges is not equivalent to the circulating pool of cells, but includes cells that are locally sequestered in the skull bones (Herisson et al., 2018; Cugurra et al., 2021). Thus an experimental design in which peripheral immune cells (e.g. monocytes) are labeled and infused may not fully capture the local cell dynamics. This limitation of the study is now stated in the revised Discussion.

In response to the second question (“Could infiltrates contribute to the transcriptomic changes through cytokine release?”), the answer is “yes”. We have tested one cytokine, CCR2, by gene KO and found that the effect of the KO on the number of leptomeningeal macrophages is modest (Figure 7). This question, and the larger question of immune cell trafficking, are now addressed in the revised Discussion section.

Cugurra et al. (2021) Skull and vertebral bone marrow are myeloid cell reservoirs for the meninges and CNS parenchyma. Science 373:eabf7844.

Herisson et al. (2018). Direct vascular channels connect skull bone marrow and the brain surface enabling myeloid cell migration. Nat Neurosci. 21:1209-1217.

6. Did collagen stain decrease upon infection, as suggested by RNAseq? The authors should also validate their markers for endothelial subtypes by IF. Also, it's unclear how veins and arteries were distinguished, and SMA stain should be used to confirm.

Collagen. We immunostained for COL14A1 in control vs infected dura. The immunostaining showed no change with 1 day of infection (we quantified the result; Author response image 1), although the mRNA level is greatly reduced, based on the RNAseq data (see the COL14A1 UMAP in Supplemental Figure 9B and the Col14a1 dot plot in Supplemental Figure 10A). Our guess is that the high stability of extracellular collagen accounts for the failure to see a change at the protein level after only one day. This is now noted in the Results section.

**Author response image 1. sa2fig1:** 

Endothelial markers. The transcripts that serve as the basis for all of the cell cluster assignments are tabulated (with references) in Supplemental Table 2. We have validated the leptomeminges vs. dura endothelial cell (EC) subtypes by immunostaining. Leptomeningesal ECs are CLDN5+,LEF1+, GLUT1+, and PLVAP-, as expected for vasculature with a blood-brain barrier (BBB), whereas dural ECs are CLDN5-, GLUT1-, and PLVAP+, as expected for non-barrier vasculature. This is in agreement with the expression of the corresponding transcripts by snRNAseq. This is now noted in the Results section. We did not see a distinct cell cluster for veins in the snRNAseq – they are presumably part of the large cluster that encompasses veins and capillaries. The artery cluster is distinct in the snRNAseq and, as described in Supplemental Table 2, it uniquely expresses known artery markers Bmx and Fbln5. Histologically, we confirmed our assignment of arteries and veins by smooth muscle actin (SMA) staining, which is abundant on arteries. This is now noted in the Results section.

7. What was the consequence of TLR4 KO, CCR2 KO and clodronate on bacterial load? Maybe 1 day post infection is too early.

Counting of RFP-expressing *E. coli*, showed no statistically significant differences in the number of *E. coli* in the leptomeninges between WT, TLR4 KO, and CCR2 KO mice at one day postinfection. Similarly, the number of *E. coli* in the leptomeninges did not differ significantly between mice that received control vs. clodronate liposomes. This data has been added to Supplemental Figure 3C. As the reviewer notes, this may reflect the fact that we are studying an acute model of meningitis at one day post-infection.

8. The discussion would need to incorporate all the data and show the big picture: are meningeal layers similarly 1/infected, 2/responsive, 3/leaky? What could be the consequence of the transcriptomic changes (collagen, slc transporters, etc.)?

Thank you for that comment. We have expanded the Discussion as indicated. At present, the physiologic significance of the many observed transcriptome changes remains to be determined.

9. What happens at later time points? Which barrier breaks the most and lets bacteria invade the tissue?

Regarding later time points, the model of acute early postnatal meningitis that we use is uniformly fatal by two days after bacterial inoculation. Therefore, we restricted our analyses to 22 hours after *E. coli* inoculation. We have now included a Kaplan-Meier survival curve as part of Supplemental Figure 3, and have expanded the Results section to more fully describe the natural history of this meningitis model. The clinical course of the mouse meningitis model parallels the natural history of untreated bacterial meningitis in humans, as seen in Figure 2 of Eisen et al. (2022) [Eisen et al. (2022)] Longer than 2 hours to antibiotics is associated with doubling of mortality in a multinational community acquired bacterial meningitis cohort. Sci Rep 12:672}. In that paper, the authors show that, in humans with bacterial meningitis, no delay from diagnosis to the start of antibiotics is associated with 10% mortality, whereas a 12 hour delay is associated with 40% mortality, with intermediate mortality values for intermediate delays in the start of antibiotic therapy. Presumably, the mortality would be even higher with no antibiotic treatment, consistent with historical accounts from the pre-antibiotic era that describe the high mortality and rapidly fatal course of bacterial meningitis.

Bacterial invasion. Current evidence points to trans-endothelial movement across the meningeal vasculature, but we have not studied that aspect of the model. For example, see Coureuil et al. (2017) A journey into the brain: insight into how bacterial pathogens cross blood-brain barriers. Nat Rev Microbiol 15:149-159.

Reviewer #2 (Recommendations for the authors):– It is not clear what the 'Ccl2+ and Ccl2-' resident macrophages are. Do they correlate to the MHCII+ and MHCII- dural macrophages? Which other markers that were previously identified by these subsets in adult mice, do they express? Are they present in both the dura and the arachnoid/pia mater? In the dura, do they localize to the sinus region? How does this compare to adult mice?

In the P6 dataset, we did not observe MHCII (e.g. H2-Aa) expression in macrophages. MHCII is expressed in a small number of monocytes (Ccr2+ cells). This differs from what has been reported for adult meningeal macrophages (Van Hove et al., 2019; see text excerpt below). We also cannot see any clear patterns among the P6 macrophages based on the other markers described by Van Hove et al. Presumably, these differences reflect the relative immaturity of meningeal macrophages at P6. This is now noted in the Discussion section. “Similarly, comparisons between P6 and adult macrophage subtypes in the mouse meninges, the latter defined by Van Hove et al. (2019), are challenging as adult meningeal macrophages were subdivided based on MHC class II (e.g., H2-Aa) transcript levels, which are uniformly low in P6 meningeal macrophages.”

Below are quotes from the relevant passages of Van Hove et al. "A single-cell atlas of mouse brain macrophages reveals unique transcriptional identities shaped by ontogeny and tissue environment." *Nature neuroscience* 22.6 (2019): 1021-1035.

“BAMs fell into two main groups: MHCIIlo and MHCIIhi, which exhibited many differentially expressed genes (for example, Mrc1, Cd163, Gas6 versus H2-Aa, Cd74, Klra2). BAMs also exhibited clear tissuespecific signatures. Signature genes were identified that specifically marked meningeal MHCIIlo BAMs (for example, Clec4n, Clec10a, Folr2) or were enriched in SD-BAMs (for example, Lyve1, P2rx7, Egfl7), individual dural BAM subsets (for example, Pla2g2d, Ccl8) or CP BAMs (for example, Lilra5, Ttr). CPepiBAMs also exhibited a unique transcriptional profile, which included expression of Cst7, Gm1673 and Clec7a.”

“BAM subsets exhibit a mixed ontogeny and a postnatal development that is highly dependent on the border region. To investigate how BAM heterogeneity develops over time, we performed flow cytometry on dissected borders starting from postnatal day 0 (Figure 7a,b). Most BAMs shared a similar phenotype at birth, being MMR^hi^ and MHCII−. While the SD-BAM pool remained relatively stable, the phenotype and composition of dural and CP-BAMs dramatically changed from postnatal day 21 onwards.”

– They mention that IL-6 was expressed by fibroblasts. However, in the image provided in Figure 2H, it cannot be concluded which cells express IL-6, apart from being negative for CD206.

We agree. The immunostaining only tells us that there are cells in the infected dura that express IL-6. The conclusion that these are fibroblasts rests on the snRNAseq data, as shown in Supplemental Figure 6A, which shows a dotplot (with IL-6 transcripts second from the left), and Supplemental Figure 8B, which shows an IL-6 UMAP plot (central pair of panels, second panels from the bottom). Thus is now stated in the Results section.

– In Figure 3, it is difficult to see the changes across immune cells in infected vs. control. Why not show a pie chart, or also UMAPs of infected vs. control side by side, or overlayed with a color (as in Figure 4A).

Figure 3D is essentially the same as what a pie chart would show – the abundances of each class of immune cell in infected and uninfected samples. Figure 3D has an advantage over a pie chart in that it plots the values from each of the three replicates for infected and uninfected samples so that one can see the reproducibility of the replicate data. The upper panel in Figure 3D plots all of the major cell types, and the lower panel plots the individual immune cell types.

– For snRNA-seq, n=? how many mice were used?

For snRNA-seq, we used 2 control (uninfected) mice and 3 infected mice, i.e. one mouse per library. This is now stated in the Results and Methods sections.

– Figure 4, does this include both meningeal compartments together?

Yes. That is now added to the figure legend.

– it is difficult to quantitively assess the frequencies of the cell populations from the sequencing data. To validate the dynamics of different cell types, validation using flow cytometry (total cell counts) would be nice to include.

It would be interesting to compare snRNAseq and FACS – something that we have not done. That said, there is good reason to think that snRNAseq is robust for quantification because the protocol is so simple: one collects the tissue, homogenizes in gentle detergent, and then collects nuclei in an iodixanol gradient by centrifugation. There is no enzymatic cell dissociation and there is minimal time involved in the tissue processing protocol relative to more involved protocols. We think that snRNAseq is especially advantageous when the cells come from a complex tissue like the meninges that includes cell types with different sizes, shapes, and adhesive properties. The isolated nuclei are probably more homogenous in their properties than are the cells from which they came. The important points to take from the data plotted in Figures 3D and 4B are not so much the absolute numbers, but the relative changes (or lack thereof) in cell type abundances in comparing uninfected and infected samples.

Our approach to cell quantification in Figures 2 and 5-8 has been to quantify cells by immunostaining of flatmounts in which the Z-stacked image spans the full thickness of the tissue. This immunostaining approach has the advantage that (1) no cells are lost or escape imaging, and (2) we see the spatial arrangements of the cells in the context of tissue architecture (this is lost in FACS).

– The authors explain the increase of macrophages in infection due to conversion of resident macrophages to inflammatory macrophages. Fate-mapping resident macrophages or monocytes prior to infection would be helpful to substantiate this claim.

The increase in macrophage numbers in the leptomeninges with infection is modest: ~20% in FVB mice and <5% with C57 (Figure 2E and Supplemental Figure 12). We have not attempted to address the movement of immune cells between compartments. We agree with the reviewer that some cells in both the leptomeninges and the dura could be recent arrivals. Our data present only a static image, and without a cell tracing analysis one cannot assess the dynamics of cell movement. We note that in the meninges a cell tracing analysis is technically non-trivial as there is good evidence that the pool of myeloid cells that supplies the meninges is not equivalent to the circulating pool of cells, but includes cells that are locally sequestered in the skull bones (Herisson et al., 2018; Cugurra et al., 2021). We have added the following sentence to the Results section to underscore this uncertainty: “Whether this increase reflects in situ proliferation, ingress from other tissue compartments [e.g. blood and/or skull bone marrow (Herisson et al., 2018; Cugurra et al., 2021)], or a combination of the two, remains to be determined.”

Herisson F, Frodermann V, Courties G, Rohde D, Sun Y, Vandoorne K, Wojtkiewicz GR, Masson GS, Vinegoni C, Kim J, Kim DE, Weissleder R, Swirski FK, Moskowitz MA, Nahrendorf M. (2018). Direct vascular channels connect skull bone marrow and the brain surface enabling myeloid cell migration. Nat Neurosci. 21:1209-1217.

Cugurra A, Mamuladze T, Rustenhoven J, Dykstra T, Beroshvili G, Greenberg ZJ, Baker W, Papadopoulos Z, Drieu A, Blackburn S, Kanamori M, Brioschi S, Herz J, Schuettpelz LG, Colonna M, Smirnov I, Kipnis J. (2021) Skull and vertebral bone marrow are myeloid cell reservoirs for the meninges and CNS parenchyma. Science 373:eabf7844.

Further support for the conclusion that relatively few macrophages have migrated into the leptomeninges wihin one day of infection comes from the data in Figure 8C and D, which show that after clodronate depletion of CNS macrophages at P3, there are very few macrophages in the leptomeninges at P6 and with infection this number increases only modestly (to ~10% of the value of untreated controls). This is now noted in the Results section describing the clodronate experiments.

– Figure 7, the difference of CLDN5 between Ccr2-/- control and infected is not evident in the images (A), yet the quantification shows a highly significant difference (B). This should be explained. In the images in (A), a WT control for both infected and non-infected should be shown.

Close inspection of Figure 7A shows that, with infection, the Ccr2 KO leptomeninges vasculature has CLDN5 clusters and also that a larger fraction of the image area is occupied by the vasculature. This is more obvious in the lower set of images (CLDN5 + Sulfo-NHS biotin) than in the upper set of images (CLDN5 + CD206). Quantification shows that the increase in area covered by blood vessels is 15-20% – not a huge difference, but statistically significant because the sample size was fairly large (we quantified >10 images for each condition).

Thank you for the suggestion to show the WT images. We have expanded Figure 7A to show them.

– Figure 7C, Tlr4-/- mice have fewer meningeal macrophages already at steady state. Can they speculate why and discuss this?

As noted by the reviewer, the uninfected Tlr4 KO has only ~70% as many leptomeningeal macrophages as does the uninfected C57 control. We don’t know why this is, and we have not explored it. One naïve thought is that postnatal pups are exposed to a low amount of LPS from their GI tract and skin microbiomes and from feces in the cage, which leads to a low level of constitutiveTLR4-dependent signaling. In Tlr4 KO mice, the absence of that “background” level of TLR4 signaling might lead to reduced macrophage proliferation. This possibility is now noted in the Results section.

– Also, in Ccr2-/- mice, fewer macrophages (CD206+ or PU.1+ cells) were found in the meninges, of both infected and non-infected mice. Is there a difference between the subsets they identified?

We have not done detailed phenotyping (e.g. snRNAseq) of the macrophages in the Ccr2 KO or Tlr4 KO mice – i.e., we did not analyze macrophage subsets in the context of these experiments. We have simply visualized and quantified them with CD206 and PU.1 immunostaining. Related to this: Supplemental Figure 13 now explicitly compares the numbers of leptomeningeal macrophages determined by counting PU.1+ cells vs. counting CD206+ cells for WT, Tlr4 KO, and Ccr2 KO. Ccr2 KO mice have ~20% fewer leptomeningeal macrophages compared to the C57 control. Given that CCR2 is the receptor for monocyte chemoattractant protein 1 (CCL2), it is not entirely surprising that a mouse lacking that receptor might have a modest reduction in macrophage abundance in the leptomeninges. This is now noted in the Results section.

– In Tlr4-/- and Ccr2-/- or clodronate-treated mice, bacteria dissemination in the meninges and brain should also be analyzed.

We have now quantified the bacterial load and the data are shown in Supplemental Figure 3. The following sentence has been added to the Results section: “At one day post-infection, the number of *E. coli* in leptomeninges flatmounts did not differ significantly between C57 WT control and Ccr2-/- mice, whereas the number of *E. coli* in leptomeninges flatmounts was increased ~2-fold in Tlr4-/- mice, albeit with substantial scatter in the data (Supplemental Figure 3C, center plot).”

– They suggest that Tlr4 is important on non-macrophages. Can they show in their snRNA-seq data which cell types express it?

Tlr4 is expressed by all of the major meningeal cell types. Ccr2 is expressed primarily by monocytes. We have added a new Supplementary Figure 12 showing this with a dot plot and UMAPs.

– It should be discussed that killing of macrophages (by clodronate) treatment, may also lead to inflammation and changes in ECs (as seen in Figure 8E).

Thank you for pointing out that possibility. There could indeed be secondary effects of clodronate treatment. For example, activation of NF-kappaB and dilation of blood vessels could be secondary inflammatory/stress responses following the large amount of macrophage cell death and the attendant release of bioactive substances. This is now noted in the Results section.

– Immunofluorescence images should be quantified throughout the manuscript. (For example, this is lacking in Figure 2F,G, Figure 5E,F).

This has now been done, with additional panels added to Figures 2, 5, 7, and 8.

– Figure 5G, for LEF, n=2, this would need to be increased to show statistical significance

Thank you for catching that. There was an error in the original figure: the LEF1 data point sitting on the line at the mean value had been inadvertently deleted. The correct image has been inserted in the revised Figure 5G – it is a comparison of 3 data points vs. 3 data points.

– Time point of analysis, repetitions and number of mice used is not always clear in the figure legends.

All analyses were conducted at P6. This was stated once in the Results section, but we have now stated it a second time (in the first and second sections within the Results section). For the snRNAseq libraries, the number of independent libraries is shown in Supplemental Table 1 and Supplemental Figure 1. The number of mice used for each image quantification is now listed in a new Supplemental Table 4. The number of repetitions for the Western blots is apparent from the data points in Figures 5D and G and is also now stated in the Figure 5 legend.